# Precise excitation-inhibition balance controls gain and timing in the hippocampus

**Aanchal Bhatia[†], Sahil Moza[†], Upinder Singh Bhalla***

National Centre for Biological Sciences, Tata Institute of Fundamental Research, Bangalore, India

**Abstract** Excitation-inhibition (EI) balance controls excitability, dynamic range, and input gating in many brain circuits. Subsets of synaptic input can be selected or 'gated' by precise modulation of finely tuned EI balance, but assessing the granularity of EI balance requires combinatorial analysis of excitatory and inhibitory inputs. Using patterned optogenetic stimulation of mouse hippocampal CA3 neurons, we show that hundreds of unique CA3 input combinations recruit excitation and inhibition with a nearly identical ratio, demonstrating precise EI balance at the hippocampus. Crucially, the delay between excitation and inhibition decreases as excitatory input increases from a few synapses to tens of synapses. This creates a dynamic millisecond-range window for postsynaptic excitation, controlling membrane depolarization amplitude and timing via subthreshold divisive normalization. We suggest that this combination of precise EI balance and dynamic EI delays forms a general mechanism for millisecond-range input gating and subthreshold gain control in feedforward networks.

**\*For correspondence:**
bhalla@ncbs.res.in

[†]These authors contributed equally to this work

## Introduction

Individual neurons in the brain can receive tens of thousands of excitatory (E) and inhibitory (I) synaptic inputs. Under normal conditions, the ratio of excitatory to inhibitory input remains invariant, a robust property of the nervous system, termed EI balance (*Anderson et al., 2000*; *Atallah and Scanziani, 2009*; *Okun and Lampl, 2008*; *Okun and Lampl, 2009*; *Wehr and Zador, 2003*). Disruption of balance is linked with several pathologies, including epilepsy, autism spectrum disorders and schizophrenia (*Yizhar et al., 2011*).

Theoretically, neurons in 'detailed balanced' EI networks receive balanced responses from all subsets of presynaptic inputs (*Vogels and Abbott, 2009*), and neurons in 'tightly balanced' EI networks receive inputs balanced at fast (<10 ms) timescales (*Denève and Machens, 2016*). Together, these properties constitute a 'precisely balanced' network (*Hennequin et al., 2017*). This precise balance on all synaptic subsets can be exploited by the brain for 'input gating'. In this process, neurons can be driven by selective shifts in EI ratios at specific inputs, while other inputs remain balanced in the background. This constitutes a flexible and instantaneous information channel local to the shifted synapses (*Kremkow et al., 2010*; *Vogels and Abbott, 2009*).

Our current understanding of EI balance is based on measurements made at single neurons in response to various stimuli. Strong EI correlations have been seen in response to series of tones in auditory cortex (*Wehr and Zador, 2003*; *Zhang et al., 2003*; *Zhou et al., 2014*), whisker stimulation in somatosensory cortex (*Wilent and Contreras, 2005*), during cortical up states in vitro (*Shu et al., 2003*) and in vivo (*Haider et al., 2006*), during gamma oscillations in vitro and in vivo (*Atallah and Scanziani, 2009*), and during spontaneous activity (*Okun and Lampl, 2008*). At the synaptic scale, the ratio of excitatory and inhibitory synapses on various dendrites of a neuron has been shown to be conserved (*Iascone et al., 2018*). However, the precision and presynaptic origin of balance is not

well understood. It remains to be established if EI balance arises transiently from complex temporal dynamics of several presynaptic layers, if it requires summation of inputs from multiple presynaptic populations, or if it exists even at subsets of a single presynaptic population. This granularity of EI balance, of both presynaptic identity and number of inputs, can determine the precision with which synaptic inputs can be selected or 'independently gated' to affect postsynaptic activity.

In this study, we address two key open questions in the field. First, can EI balance arise even in a single layer feedforward network, and if so, at what granularity of network subsets do postsynaptic cells experience balanced excitation and inhibition? Second, how do excitation and inhibition integrate to encode and communicate information at the postsynaptic neuron? We addressed these questions in vitro, to isolate the hippocampal network from background activity, and to deliver precisely controlled combinatorial stimuli. We stimulated channelrhodopsin-2 (ChR2) expressing CA3 neurons in several combinations using optical patterns, and measured responses in CA1.

We report that hundreds of randomly chosen subsets of CA3 neurons provide excitatory and feedforward inhibitory inputs to CA1 cells with a close to identical ratio, demonstrating for the first time, precise balance (*Hennequin et al., 2017*) in the brain. On examining the integration of excitation and feedforward inhibition, we found that inhibition arrives with a dynamically varying onset delay that decreases with increasing input amplitude. This leads to a characteristic initial linear portion in the neuronal input-output curve where the inhibition arrives too late to affect peak depolarization, and a progressively diminishing output as the EI delay decreases with increasing input. This novel gain control operation, termed Subthreshold Divisive Normalization (SDN) encodes input information in both amplitude and timing of the CA1 response.

## Results

In our study, we first utilize and characterize an optical stimulation protocol for CA3 pyramidal neurons, and measure intracellular responses at CA1 pyramidal neurons (*Figure 1*). We then demonstrate precise EI balance for various combinations of CA3 inputs at CA1 using voltage clamp to separate the E and I components (*Figure 2*). Next, we measure the depolarization at CA1 due to summation of E and I using different input combinations (*Figure 3*), and show sublinearity of summation. Expansion of the range of inputs revealed divisive normalization and suggested that another factor such as inhibitory kinetics should be included to account for the sublinearity (*Figure 4*). In *Figure 5*, we confirm that blocking inhibition leads to much reduced sublinearity of summation, and that inhibition scales linearly with stimulus amplitude. We then establish that inhibitory delay is crucial for explaining the sublinearity in SDN (*Figure 6*). In *Figure 7*, we show that post-synaptic potential peak amplitude and timing both carry information about the summed stimulus amplitude, and show that this information carries over to spike timing. In *Figure 8*, we summarize the analysis and suggest how SDN could contribute to input gating in the hippocampus.

### Optical stimuli at CA3 elicit subthreshold responses at CA1

To provide a wide range of non-overlapping stimuli, we projected patterned optical stimuli onto channelrhodopsin-2 (ChR2) expressing CA3 neurons in acute hippocampal slices. We used CA3-cre mice to achieve CA3-specific localization of ChR2 upon injection of a Lox-ChR2 virus (*Figure 1a*, Materials and methods). We used a Digital Micromirror Device (DMD) projector (Materials and methods, *Figure 1—figure supplement 1*) to generate spatiotemporal optical patterns in the form of a grid of several 16 um x 16 um squares, each square approximating the size of a CA3 soma (*Ishizuka et al., 1995*) (*Figure 1d*). This grid was centered at the CA3 cell body layer, and extended to the dendritic layer (*Figure 1a*). Each optical pattern consisted of 1 to 9 such randomly chosen grid squares, presented to CA3 cells as stimulus, at an inter-stimulus interval of 3 s (*Figure 1a,d*, Materials and methods). In a typical experiment, several randomly chosen stimulus patterns with different number of input squares were delivered to CA3, in three successive repeats. We first characterized how CA3 responded to the grid stimulation (*Figure 1b,e,f,g*). CA3 neurons fired reliably with a < 2 ms jitter, calculated as the standard deviation of the time of first spike (*Figure 1f*) (n = 8 CA3 cells, inputs = 52, median = 0.44 ms, N = 1 to 9 squares). No desensitization occurred during the timeframe of an experiment, and the probability of spiking remained constant between the three repeats (*Figure 1g*) (n = 7 CA3 cells, N = 1 to 9 squares). Thus, we could stimulate CA3 with hundreds of distinct optical stimuli in each experiment.

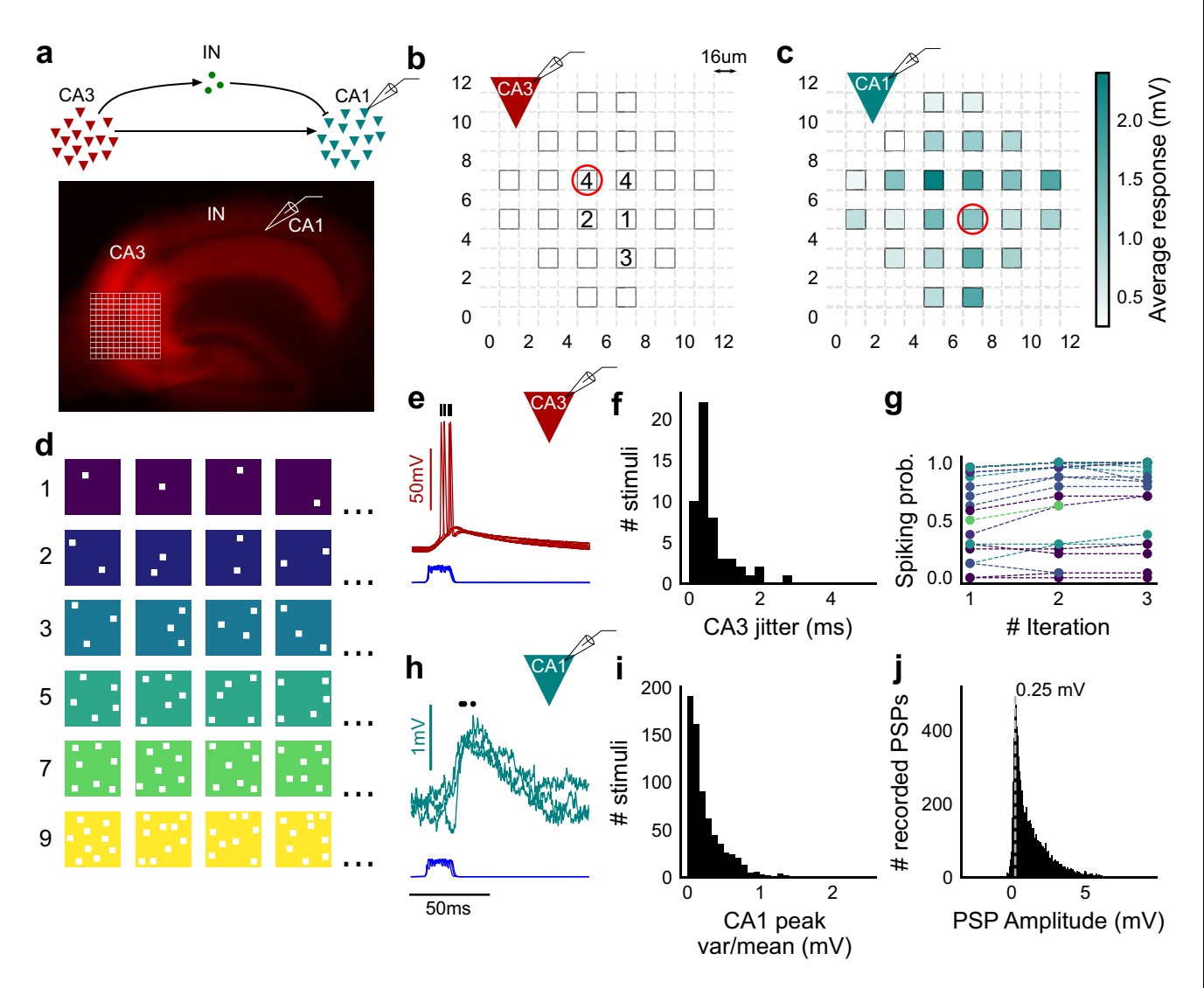

**Figure 1.** Stimulating CA3-CA1 network with hundreds of optical patterns. (a) Top, schematic of the CA3-CA1 circuit with direct excitation and feedforward inhibition. Bottom, image of a hippocampus slice expressing ChR2-tdTomato (red) in CA3 in a Cre-dependent manner. Optical stimulation grid (not drawn to scale) was centered at the CA3 cell body layer and CA1 neurons were patched. (b) Spike response map of CA3 neuron responses with one grid square active at a time. A CA3 neuron was patched and optically stimulated, in random spatio-temporal order, on the grid locations marked with grey border. This cell spiked (marked with number inside representing spike counts over four trials) in 5 out of 24 such one square stimuli delivered. (c) Heatmap of CA1 responses while CA3 neurons were stimulated with one square optical stimuli. Colormap represents peak $V_m$ change averaged over three repeats. (d) Schematic of optical stimulus patterns. Examples of combinations of N-square stimuli where N could be 1, 2, 3, 5, 7 or 9 (in rows). (e) Spikes in response to four repeats for the circled square, in b. Spike times are marked with a black tick, showing variability in evoked peak times. Blue trace at the bottom represents photodiode measurement of the stimulus duration. Scale bar for time, same as h. (f) Distribution of jitter in spike timing (SD) for all stimuli for all CA3 cells (n = 8 cells). (g) CA3 spiking probability (fraction of times a neuron spiked across 24 stimuli, repeated thrice) is consistent over a single recording session. Randomization of the stimulus pattern prevented ChR2 desensitization. Circles, colored as in d depict spiking probability on each repeat of a stimulus set with connecting lines tracking three repeats of the set (n = 7 cells). (h) PSPs in response to three repeats of the circled square in c. Peak times are marked with an asterisk. Blue traces at the bottom represent corresponding photodiode traces for the stimulus duration. (i) Distribution of peak PSP amplitude variability (variance/mean) for all 1-square responses (n = 28 cells, stimuli = 695). (j) Histogram of peak amplitudes of all PSPs elicited by all 1-square stimuli, over all CA1 cells. Grey dotted line marks the mode (n = 38 cells, trials = 8845).

The online version of this article includes the following figure supplement(s) for figure 1:

**Figure supplement 1.** Experiment design.

We then recorded postsynaptic potentials (PSPs) evoked at patched CA1 neurons while optically stimulating CA3 cells (*Figure 1c,h,i,j*). A wide range of stimulus positions in CA3 excited CA1 neurons (*Figure 1c*). Stimulation of CA3 elicited excitation and feedforward inhibition at CA1 (*Figure 1a*, *Figure 2*). Most stimuli elicited subthreshold responses (N = 1 to 9 squares). Action potentials occurred in only 0.98% of trials (183 out of 18,668 trials, n = 38 cells, N = 1 to 9 squares). This helped rule out any significant feedback inhibition from CA1 interneurons for all our experiments. Restriction of ChR2 to CA3 pyramidal cells, coupled with the fact that ~99% of all recorded CA1 responses were subthreshold, ensured that the recorded inhibition was largely feedforward (disynaptic) (*Figure 1a*). Responses to the same 1-square stimulus were consistent, 84.74% responses showed less than 0.5 variance by mean (695 stimuli, three repeats each, n = 28 cells, N = 1 square) (*Figure 1i*). Notably, the distribution of all one square responses had a mode at 0.25 mV, which is close to previous reports of a 0.2 mV somatic response of single synapses in CA1 neurons (*Magee and Cook, 2000*) (8845 trials, n = 38 cells, N = 1 square) (*Figure 1j*).

## Arbitrarily chosen CA3 inputs show precise EI balance at CA1

To examine the relationship between excitation and inhibition, we voltage clamped CA1 neurons, first at the inhibitory (−70 mV) and then at the excitatory (0 mV) reversal potential to record Excitatory and Inhibitory Post Synaptic Currents (EPSCs and IPSCs) respectively. We first presented five different patterns of 5 squares each, at both of these potentials, and recorded EPSCs and IPSCs. We found strong proportionality between excitation and inhibition for every stimulus pattern (*Figures 1d* and *2a*). This suggested that inputs from even random groups of CA3 neurons may be balanced at CA1. Repeats with the same stimulus pattern gave consistent responses, but different patterns evoked different responses (*Figure 2a*, *Figure 2—figure supplement 1b*). This indicated that the optically driven stimuli were able to reliably activate different subsets of synaptic inputs on the target neuron. Next, we asked, in what range of input strengths does random input yield balance? We presented five different patterns for each of 1, 2, 3, 5, 7 or 9 square combinations at both recording potentials. Surprisingly, all stimuli to a cell elicited excitatory and inhibitory responses in the same ratio, irrespective of response amplitude (*Figure 2b,c*) (n = 13 CA1 cells, area under curve, mean $R^2$ = 0.89 +/- 0.06 SD, *Figure 2—figure supplement 2*). Notably, the mode of single-square responses was ~0.25 mV, close to single synapse PSP estimates (*Magee and Cook, 2000*) (*Figure 1j*). However, accounting for the low (~0.2) release probabilities ($P_r$) at the CA3-CA1 synapse (*Murthy et al., 1997*), we should be able to see a single synapse response if approximately $1/P_r$ synapses were activated. Hence, we estimate that the granularity of the balance as resolved by our method is of the order of 5–10 synapses (*Figure 2—figure supplement 1d,e*). The slope of the regression line through all stimulus-averaged responses for a CA1 cell was used to calculate the Inhibition/Excitation (I/E) ratio for the cell. IPSC/EPSC ratio was typically between 2 and 5 (*Figure 2d*). The variability of I/E ratios over all stimuli for a cell was lower than the variability of all stimuli across cells (for 12 out of 13 cells, *Figure 2—figure supplement 1c*). The high $R^2$ values for all cells showed tight proportionality for all stimuli (*Figure 2e*). The residual distribution remained symmetric for increasing numbers of spots, again showing that they were not affected by the number of stimulus squares presented (*Figure 2—figure supplement 1a*). While feedforward inhibition is expected to increase with excitation, convergence of I/E ratios for randomly chosen inputs to a cell to a single number was unexpected, since shared interneurons consist of only about 10% of the total neuronal population (*Woodson et al., 1989*; *Bezaire and Soltesz, 2013*).

## Detailed balance requires co-tuning of EI weights

We next tested the hypothesis that the observed correlation between excitatory and inhibitory inputs was due to an averaged sum over many untuned (globally balanced) synapses, as opposed to a much finer granularity of tuning between excitatory and inhibitory synaptic weights (detailed balance). To address this, we modelled excitatory and inhibitory synaptic weights to a neuron with different amounts of weight tuning, parameterized by rho ($\rho$), which takes values between 0 (no tuning or global balance) and 1 (detailed balance) (Materials and methods, *Figure 2—figure supplement 1h*). For values between 0 and 1, $\rho$ determined the degree of correlation between the basal excitatory and inhibitory synaptic weights. To test if weight tuning was necessary to observe balance, we modeled the summation of synaptic inputs with the premise that excitatory and inhibitory afferents

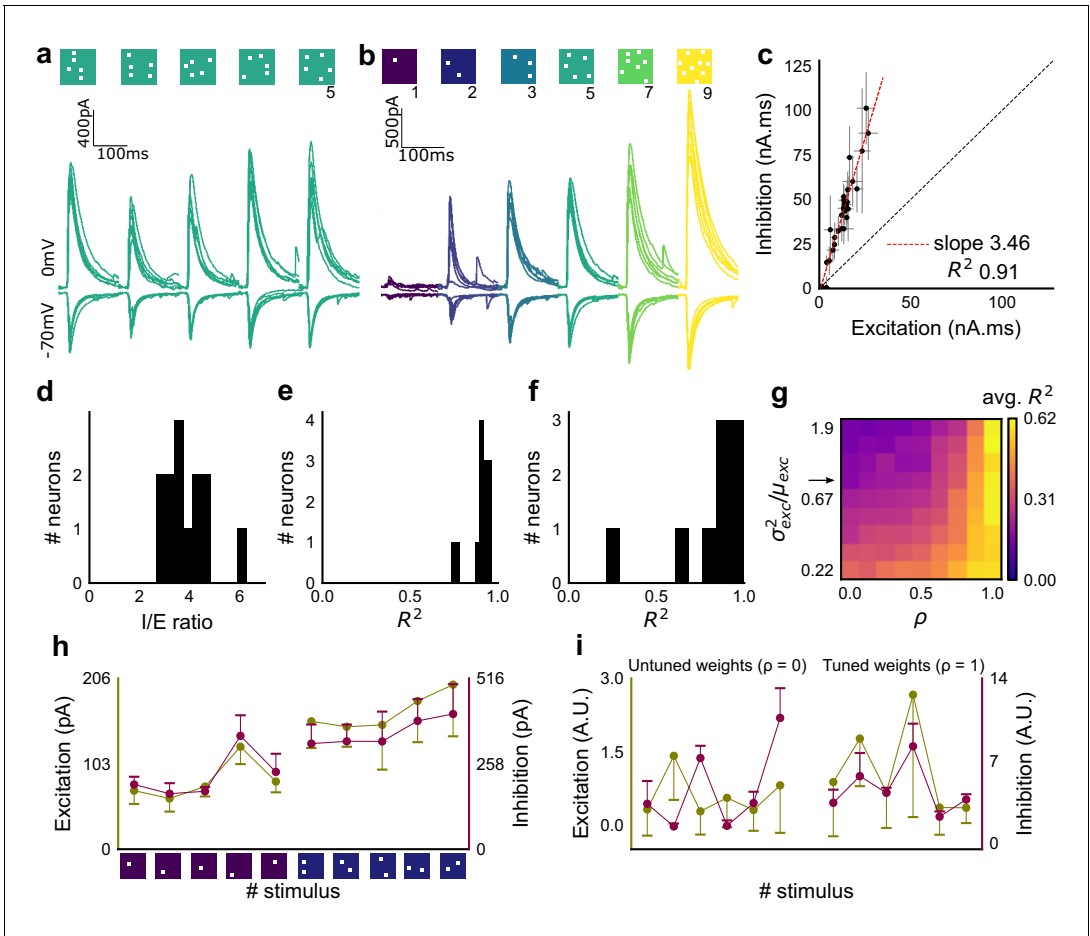

**Figure 2.** Excitation and inhibition are tightly balanced for all stimuli to a CA1 cell. (a) Monosynaptic excitatory postsynaptic currents (EPSCs, at −70 mV) and disynaptic inhibitory postsynaptic currents (IPSCs, at 0 mV) in response to five different stimulus combinations of 5 squares each. All combinations show proportional excitatory and inhibitory currents over six repeats. Top, schematic of 5-square stimuli. (b) EPSCs and IPSCs are elicited with the same I/E ratio in response to six repeats of a combination, and across six different stimuli from 1 square to nine squares, for the same cell as in a. Top, schematic of the stimuli. (c) Area under the curve for EPSC and IPSC responses, obtained by averaging over six repeats, plotted against each other for all stimuli to the cell in a, b. Error bars are s.d. (d) Summary of I/E ratios for all cells (n = 13 cells). (e) Summary for all cells of R² values of linear regression fits through all points. Note that 11 out of 13 cells had R² greater than 0.9, implying strong proportionality. (f) Same as e, but with linear regression fits for 1 and 2 square responses, showing that even small number of synapses are balanced for excitation and inhibition (n = 9 cells). (g) Phase plot from the model showing how tuning of synapses (ρ) affects observation of EI balance (R²) for various values of variance/mean of the basal weight distribution. Changing the scale of the basal synaptic weight distributions against tuning parameter ρ affects goodness of EI balance fits. Arrow indicates where our observed synaptic weight distribution lay. (h) Example of EI correlations (from data) for 1 and 2 square inputs for an example cell. Bottom, schematic of the stimuli. Excitation and inhibition are colored olive and purple, respectively. Error bars are s.d. (i) Examples of EI correlation (from model) for small number of synapses, from the row marked with arrow in g. The left and right curves show low and high correlations in mean amplitude when EI synapses are untuned (ρ = 0) and tuned respectively (ρ = 1) (A.U. = Arbitrary Units). Colors, same as h. Error bars are s.d.

The online version of this article includes the following figure supplement(s) for figure 2:

**Figure supplement 1.** Detailed balance in CA3-CA1 feedforward network.

**Figure supplement 2.** Raw data from all cells showing precise balance between excitation and inhibition.

will be activated strictly proportionally (number balance). We then tested how mean and variance correlations between EI amplitudes changed with different degrees of weight tuning.

We observed tight correlations between EI inputs without weight tuning, but only if the basal synaptic weight distribution was narrow. Further, for a narrow weight distribution, the change from global to detailed balance had little effect on mean EI amplitude correlations. In contrast, weight tuning was required to see EI balance for wider synaptic weight distributions, especially for stimuli which activated small numbers of synapses (*Figure 2g*). We next calculated the width of the smallest responses (1-square GABAzine EPSP) as a proxy for the basal weight distribution (*Figure 3—figure*

*supplement 1a*). The observed responses were broadly distributed. With this basal weight distribution, the model exhibited EI balance only when the excitatory and inhibitory synaptic weights were co-tuned, that is, maintained at the same ratio (marked with arrow in *Figure 2g*, Materials and methods, *Figure 2—figure supplement 1h*).

With the reasoning developed above, we checked for EI balance in the smallest inputs in our datasets - 1 and 2 square data from voltage clamped cells (having five or more input patterns per cell) (*Figure 2—figure supplement 1d*), and only one square from current clamped cells (24 inputs per cell) (*Figure 2—figure supplement 1e,f*). We found that the responses corresponding to a few synapses per input were balanced (*Figures 1j* and *2f,h*, *Figure 2—figure supplement 1d,e,f*), suggesting tuning of excitatory and inhibitory weights.

In addition, the model also predicted a tuning dependent change in the correlations of variability of excitation and inhibition amplitudes for repeats of the same stimulus. For a wide synaptic weight distribution, increase in tuning increased EI variability correlations (*Figure 2—figure supplement 1i, k*). As with EI mean correlations (*Figure 2h,i*), weight tuning had little effect in the case of narrow synaptic weight distributions. Again, our calculated synaptic weight distribution was in the range where strong variability correlations would be seen only if synaptic weights were tuned. We found strong correlations between excitatory and inhibitory standard deviation between six repeats of the same stimulus in our voltage-clamp dataset, suggesting that there is detailed balance in the network (*Figure 2—figure supplement 1g,j*).

Thus, we present three observations using small (one and two square) stimulus strengths: a wide basal weight distribution, correlated mean EI amplitude and correlated EI amplitude variability. Together, these are inconsistent with the hypothesis that EI balance can emerge with no other requirement than a proportional increase in number of EI afferents in a globally balanced network. This supports the existence of weight tuning and hence detailed balance in the CA3-CA1 network.

Overall, we found stimulus-invariant proportionality of excitation and inhibition for any randomly selected input, over a large range of stimulus strengths from a single presynaptic network. In addition to detailed balance, we show below that there is tight balance, that is the timing of the balanced feedforward inhibition was within a few milliseconds of the excitation (*Figure 6g,h*). Thus, we concluded that the CA3-CA1 circuit exhibits precise (both detailed and tight) balance (*Hennequin et al., 2017*).

## Combinatorial CA3 inputs sum sublinearly at CA1

We next asked how CA3 inputs, that lead to balanced excitatory and feedforward inhibitory conductances, transform into membrane potential change at CA1 neurons. Based on anatomical studies, CA3 projections are likely to arrive in a distributed manner over a wide region of the dendritic tree of CA1 pyramidal neuron (*Ishizuka et al., 1990*) (*Figure 3a*). While pairwise summation at CA1 has been shown to be largely linear in absence of inhibition (*Cash and Yuste, 1999*), the degree of heterogeneity of summation in response to spatially distributed excitatory and inhibitory synaptic inputs is not well understood (except, see *Lovett-Barron et al., 2012*) . To avoid biases that may arise from a single response measure during input integration (*Poirazi et al., 2003*), we examined PSPs using four different measures (*Figure 3c*). These were peak amplitude, area under curve (AUC), average membrane potential and area under curve till peak (*Figure 3c*).

We looked at input integration by presenting stimulus sets of 5 input squares to a given cell, with each stimulus set ranging from 24 to 225 combinations of inputs. We initially tested the center of our range of 1–9 squares (5-square inputs) before expanding the dataset to the full range (*Figure 4*). We also recorded the responses to all squares of the grid individually (one square input). The one square PSP peak response amplitude with inhibition intact (control) was not distinguishable from that with inhibition blocked (GABAzine) (Materials and methods, *Figure 3—figure supplement 1a*). As analyzed below (*Figure 6*), we find that the apparent lack of effect of GABAzine for very small inputs is because inhibition arrives with a delay that does not affect the peak response of the neuron (*Video 1*). Since individual neurons may be targeted by more than one grid square (*Figure 1b*), individual spots are not completely independent and may interact, especially given the spread in the CA3 pyramidal neuronal arbour. Our analyses show that this interaction does not have a strong or unidirectional effect on the responses of the combinations of squares (*Figure 4—figure supplement 1*, *Figure 5b,d*). The 'observed' response for a given square combination was plotted against the 'expected' response, obtained by linearly summing 1-square responses constituting that

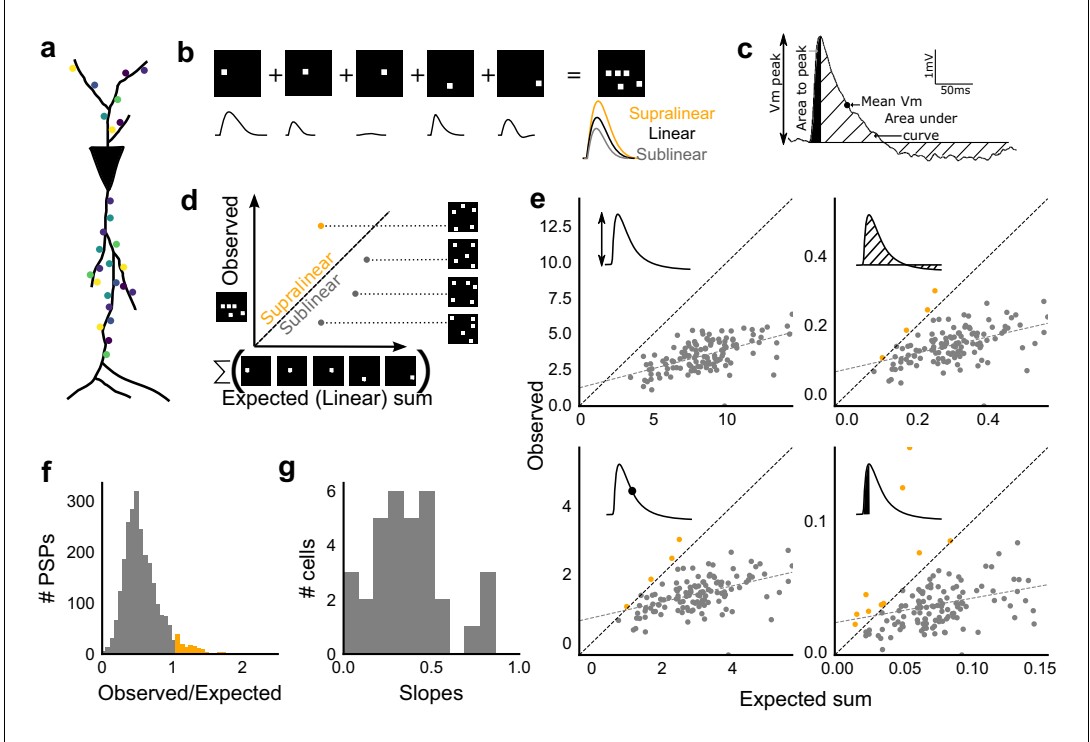

**Figure 3.** Excitatory and feed-forward inhibitory inputs from CA3 integrate sublinearly at CA1. (a) Schematic of a neuron receiving synaptic input distributed over its dendritic tree. (b) Schematic of input integration. Top, five 1-square stimuli presented individually, and a single 5-square stimulus comprising of the same squares. Bottom, PSPs elicited as a response to these stimuli. 5-square PSP can be larger (supralinear, orange), equal to (linear, black), or smaller (sublinear, grey) than the sum of the single square PSPs. (c) A PSP trace marked with the four measures used for further calculations. PSP peak, PSP area, area to peak and mean voltage are indicated. (d) Schematic of the input integration plot. Each circle represents response to one stimulus combination. 'Observed' (true response of 5 square stimulation) on Y-axis and 'Expected' (linear sum of 1 square responses) is on X-axis. (e) Most responses for a given cell show sublinear summation for a 5-square stimulus. The four panels show sublinear responses for four different measures (mentioned in c) for the same cell. The grey dotted line is the regression line and the slope of the line is the scaling factor for the responses for that cell. For peak (mV), area (mV.ms), average (mV), and area to peak (mV.ms); slope = 0.27, 0.23, 0.23, 0.18; $R^2$ 0.57, 0.46, 0.46, 0.26, respectively. The responses to AUC and average are similar because of the similarity in the nature of the measure. (f) Distribution of Observed/Expected ratio of peaks of all responses for all 5-square stimuli (mean = 0.57, s.d. = 0.31), from all recorded cells pooled. 93.35% responses to 5-square stimuli were sublinear (2513 PSPs, n = 33 cells). (g) Distribution of slopes for peak amplitude of 5-square stimuli (mean = 0.38, s.d. = 0.22). Regression lines for all cells show that all cells display sublinear (<1 slope) summation (n = 33 cells).

The online version of this article includes the following figure supplement(s) for figure 3:

**Figure supplement 1.** Summation at CA3-CA1 network is sublinear.

combination (*Figure 3b,d*). Perfectly linear summation would imply that a multi-square combination of inputs would elicit the same response as the sum of the responses to the individual squares (dotted line, *Figure 3d*). *Figure 3e* shows responses of a single cell stimulated with 126 distinct 5-square combinations. The 'observed' response was sublinear as compared to the 'expected' summed response, for most stimuli (*Figure 3e*). For all the four measures in *Figure 3c*, CA3 inputs summed sublinearly at CA1 (*Figure 3e,g*, *Figure 3—figure supplement 1c*). At this point, we hypothesised that the observed sublinearity might mostly be due to inhibition divisively scaling excitation, since excitatory and inhibitory conductances were proportional for all stimuli (*Figure 2*). We later tested this hypothesis by blocking inhibition (*Figure 5*). For all responses measured over all cells, 93.35% responses were individually sublinear, with distribution having mean 0.57 ± 0.31 (SD) (*Figure 3f*, *Figure 3—figure supplement 1d*). The slope of the regression line, which indicated the extent of sublinearity, varied between cells, with mean 0.38 ± 0.22 (SD) (n = 33 cells) (*Figure 3g*).

Thus, we found that the CA3-CA1 network exhibits sublinear summation over a large number of inputs.

## CA3-CA1 network performs Subthreshold Divisive Normalization

We then tested how summation sublinearity scaled with a larger range of inputs. We noted that nonlinear functions can be observed better with a large range of inputs (*Poirazi et al., 2003*), and therefore increased the stimulus range (*Figure 4—figure supplement 2*). GABAergic inhibition has been shown to be responsible for sublinear summation when Schaffer collateral and perforant path inputs are delivered simultaneously to CA1 (*Enoki et al., 2001*). We hypothesized that the sublinearity within the CA3-CA1 network might also occur due to the effect of inhibition. In general, inhibition may interact with excitation to perform arithmetic operations like subtraction, division, and normalization (*Carandini and Heeger, 2011*). In order to predict the operation performed by EI integration at the CA3-CA1 network, we created a composite phenomenological model to fit and test for the above three possibilities: subtractive inhibition, divisive inhibition, and divisive normalization (*Equation (1)*). We later address the mechanism using a biophysical model (*Figure 6*). *Equation (1)* describes how inhibition controls the 'observed' response ($\theta$) as a function of 'expected' response ($\varepsilon$), for the above three operations. Alpha ($\alpha$) can be thought to be a subtractive inhibition parameter, beta ($\beta$) as a divisive inhibition parameter, and gamma ($\gamma$) a normalization parameter (*Figure 4a*).

$$\theta = \varepsilon - \frac{\beta\varepsilon}{\gamma + \varepsilon}\varepsilon - \alpha \tag{1}$$

Using the framework of *Equation (1)*, we asked what computation was performed at the CA3-CA1 network. We recorded from CA1 cells while stimulating CA3 with many combinations of 2, 3, 5, 7 or 9 squares (*Figure 4b*). We selected cells with at least 50 input combinations, and pooled responses from all stimuli to a cell. Then, we fit *Equation (1)* to the PSP amplitudes (*Figure 4b*). From visual inspection, the subtractive inhibition model, $\theta = \varepsilon - \alpha$ (fixing $\beta$, $\gamma = 0$) was a bad fit, since intercepts ($\alpha$) were close to 0 (*Figure 4a*).

By fixing $\gamma$ and $\alpha$ to 0 in *Equation (1)* we obtained the Divisive Inhibition (DI) model. In this form, $\beta$ can be thought of as I/E ratio. Increasing $\beta$ decreases the observed response ($\theta$) (*Figure 4a*).

$$\theta = \varepsilon - \beta\varepsilon \tag{2}$$

Similarly, $\beta$ was fixed to 1 and $\alpha$ to 0 to get the Divisive Normalization (DN) model. This form of the equation was inspired by the analogous canonical divisive normalization equation for firing rates (*Carandini and Heeger, 2011*). Here, decrease in $\gamma$ implies increase in normalization (*Figure 4a*).

$$\theta = \varepsilon - \frac{\varepsilon}{\gamma + \varepsilon}\varepsilon = \frac{\gamma\varepsilon}{\gamma + \varepsilon} \tag{3}$$

We used least-squares polynomial regression to fit DI and DN models to our data. The goodness of fit for all cells was tested by comparing BIC (Bayesian Information Criterion) (*Figure 4c*) and reduced chi-squares of the models (*Figure 4—figure supplement 2o*, Materials and methods). DN ($\alpha = 0$, $\beta = 1$) was better than DI ($\alpha = 0$, $\gamma = 0$) model in explaining the data (BIC: Two-tailed paired t-test, p<0.00005, reduced chi-square: Two-tailed paired t-test, p<0.00005, n = 32 cells).

Subthreshold Divisive Normalization (SDN) can be clearly seen in *Figure 4b*, where observed responses to stimuli with 5 mV and 15 mV expected responses are very similar. This shows that SDN allows CA1 cells to integrate a large range of inputs before reaching spike threshold. Thus, testing with a larger range of inputs showed that the initial finding of constant I/E ratios from *Figure 2* needed to be elaborated based on the observed response saturation with increasing input strength. Potential mechanisms for this could be nonlinear summation of excitation and inhibition at the soma (tested in *Figure 5*) and inhibitory delays (examined in *Figure 6*). In summary, we observed SDN as an outcome of integration of precisely balanced inputs in the CA3-CA1 network.

## CA3 feedforward inhibition is necessary for SDN

We first verified our hypothesis that SDN results from feedforward inhibition in the CA3-CA1 network, and not from intrinsic properties of the CA1 neuron. We thus blocked inhibition and repeated the above experiment. We expected that SDN would be lost and linearity would be reinstated upon blocking inhibition.

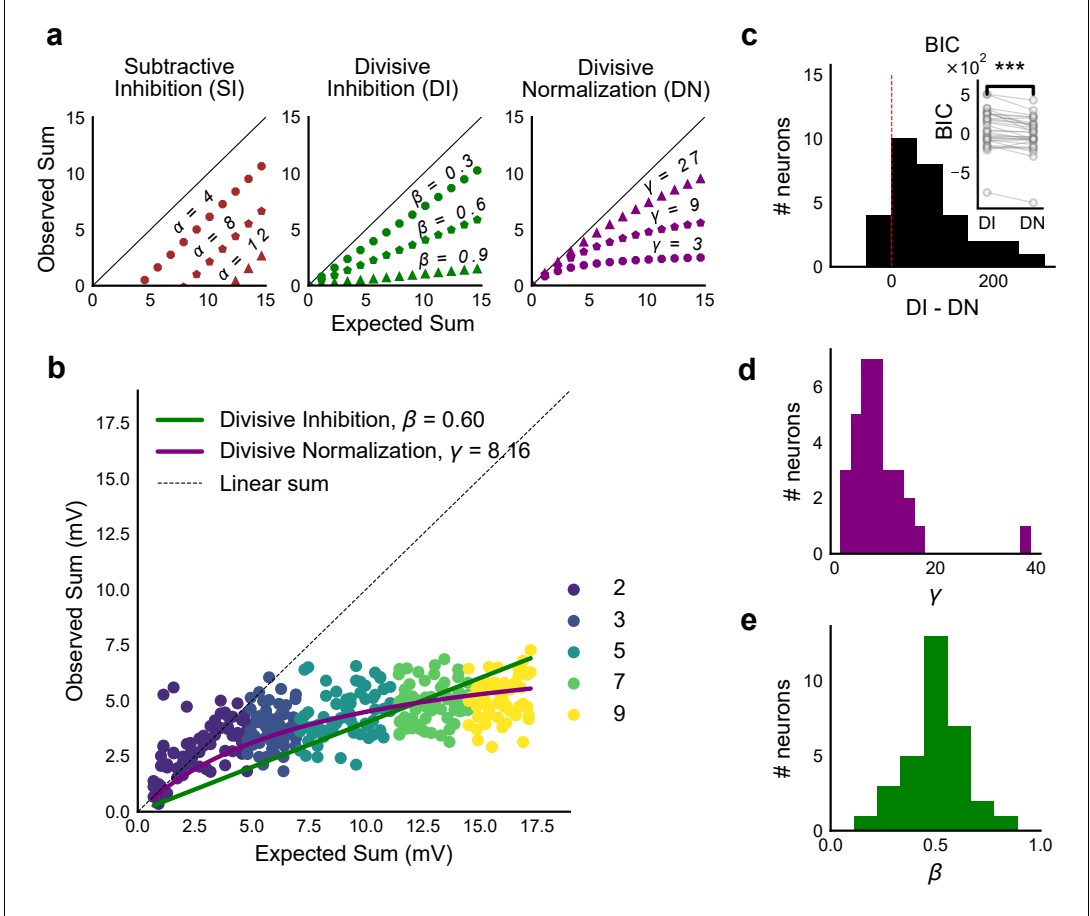

**Figure 4.** Over a wide input range, integration of CA3 excitatory and feed-forward inhibitory input leads to SDN at CA1. (a) Three phenomenological models of how inhibition interacts with excitation and modulates membrane potential: (left to right) Subtractive Inhibition (SI), Divisive Inhibition (DI) and Divisive Normalization (DN). Note how parameters $\alpha$, $\beta$ and $\gamma$ from *Equation (1)* affect response output. (b) Divisive normalization seen in a cell stimulated with 2, 3, 5, 7 and 9 square combinations. DN and DI model fits are shown in purple and green, respectively. (c) Difference in Bayesian Information Criterion (BIC) values for the two models - DI and DN. Most differences between BIC for DI and DN were less than 0, which implied that DN model fit better, accounting for the number of variables used. Insets show raw BIC values. Raw BIC values were consistently lower for DN model, indicating better fit (Two-tailed paired t-test, p<0.00005, n = 32 cells). (d) Distribution of the parameter $\gamma$ of the DN fit for all cells (median = 7.9, n = 32 cells). Compare with a, b to observe the extent of normalization. (e) Distribution of the parameter beta of the DI fit for all cells (mean = 0.5, n = 32 cells). Values are less than 1, indicating sublinear behaviour.

The online version of this article includes the following figure supplement(s) for figure 4:

**Figure supplement 1.** Interaction of squares does not affect summation unidirectionally.

**Figure supplement 2.** Input range expansion for observing nonlinear summation and divisive normalization.

We recorded responses of CA1 cells to our array of optical stimuli (*Figures 1d* and *5a*), then applied GABAzine to the bath and repeated the stimulus array (*Figure 5b*). We found that when inhibition was blocked, summation approached linearity (*Figure 5b,c*). We compared the scaling parameter $\gamma$ of the divisive normalization model fit, for the above two conditions (*Equation (3)*). The values of $\gamma$ were larger with inhibition blocked, indicative of approach to linearity (Wilcoxon rank-sum test, p<0.05, n = 8 cells) (*Figure 5c*). While inhibition accounted in large part for the observed sublinear summation, the cells with inhibition blocked showed some residual sublinearity at high stimulus levels, which has been previously attributed to $I_A$ conductance in CA1 neurons (*Cash and Yuste, 1999*). Based on the conductance equation (*Equation (5)*), leak conductance also contributes in part to the residual sublinearity (Supplementary *Equations (6-8)*). Thus, we confirmed that blocking inhibition reduced sublinearity, attenuating SDN.

## Precise balance is also seen at resting membrane potential

Then, we hypothesised that the membrane potential change evoked by inhibitory synaptic currents could be increasing non-linearly with increasing CA3 input, even though the I/E ratio of conductances would be consistent across the range of input strengths. To address this, we compared responses to identical patterns before and after GABAzine application. For a given cell, for each pattern, we subtracted the initial control response with inhibition intact from the corresponding response with inhibition blocked. This gave us the inhibitory component or 'derived inhibition' for each stimulus pattern (*Figure 5d*, inset). We found that all stimuli to a cell evoked proportional excitation and inhibition even when recorded at resting potential (*Figure 5d,e*). Thus, we rejected our hypothesis of non-linear increase in inhibitory post-synaptic potential (IPSP) with excitatory post-synaptic potential at resting membrane potential (EPSP). Over the population, the median slope of the proportionality line was around 0.7, indicating that the EI balance was slightly tilted towards higher excitation than inhibition (*Figure 5f*). IPSP/EPSP ratios (*Figure 5f*) were smaller than IPSC/EPSC ratios (*Figure 2d*) due to proximity of inhibition to its reversal ($\sim -70$ mV), than excitation to its reversal (~0 mV), at resting membrane potential ($\sim -65$ mV). Overall, we saw precise balance in evoked excitatory and inhibitory synaptic potentials for >100 combinations per neuron.

## Advancing inhibitory onset with increasing input explains SDN

We made a single compartment conductance model (*Figure 6—figure supplement 1a*, *Equation (5)*) to analyze the mechanism of SDN. We first show a Hodgkin-Huxley (HH) type single compartment model (Materials and methods), where we have used data from our voltage clamp recordings (*Figure 2*), as input to the model. Simulation with both excitation and inhibition produced curves resembling SDN, while only excitation gave a more linear response (*Figure 6a*, *Figure 6—figure supplement 2*), hence reproducing the observations depicted in *Figure 5*. Again, fit parameter $\gamma$ was significantly higher for the cases without inhibition (*Figure 6—figure supplement 2*, Wilcoxon rank sum test, p<1e-4, n = 13). In order to dissect the mechanism, we wanted to have finer control over synaptic input parameters like kinetics and EI delay.

With this in mind, we fit a function of difference of exponentials (Materials and methods) to our voltage clamp data to extract the peak amplitudes and kinetics of excitation and inhibition currents (Materials and methods). We used these and other parameters from literature (*Supplementary files 1* and *2*), and constrained the model to have EI balance, that is have maximum excitatory ($g_{\mathrm{exc}}$) and inhibitory conductance ($g_{\mathrm{inh}}$) proportional to each other, with a given I/E ratio. To test for SDN, we simulated our model in the range of experimentally determined I/E ratios, ranging from 1 to 6.

We observed that EI balance with static EI delay led to a slightly sublinear response which can be approximated with a divisive inhibition model (*Figure 6*). In contrast, subthreshold divisive normalization (SDN) implies progressively smaller changes in peak PSP amplitude with increase in excitatory input. We surmised that without changing EI balance, SDN should result if the inhibitory onset delays were an inverse function of the excitation (*Figure 6e*, *Equation (4)*). Hence, we simulated the model with dynamic delay, that is with values of inhibitory delay ($\delta_{\mathrm{inh}}$) varying as a decreasing function of the excitation.

$$\delta_{\mathrm{inh}} = \delta_{\mathrm{min}} + m e^{-k g_{\mathrm{exc}}} \tag{4}$$

Here, $\delta_{\mathrm{min}}$ is the minimum synaptic delay between excitation and inhibition, k sets the steepness of the delay change with excitation, and $m$ determines the maximum synaptic delay. In *Figure 6c*, we show that SDN emerged when we incorporated delays changing as a function of the total excitatory input to the model neuron.

We then tested this model prediction. From the EPSC and IPSC curves, we extracted excitatory and inhibitory onsets (Materials and methods), and subtracted the average inhibitory onsets from average excitatory onsets to get inhibitory delay ($\delta_{\mathrm{inh}}$) for each stimulus combination. We saw that $\delta_{\mathrm{inh}}$ indeed varied inversely with total excitation ($g_{\mathrm{exc}}$) (*Figure 6f,g*). Notably, the relationship of delay with conductance, with data from all cells pooled, seems to be a single inverse function, and might be a network property (*Figure 6g*, *Figure 6—figure supplement 1d*). The input-dependent change in inhibitory delay could be attributed to delayed spiking of interneurons with small excitatory inputs, and quicker firing with larger excitatory inputs. We further illustrate that this delay function emerges naturally by simply applying a threshold to the rising curve of an EPSP at an interneuron

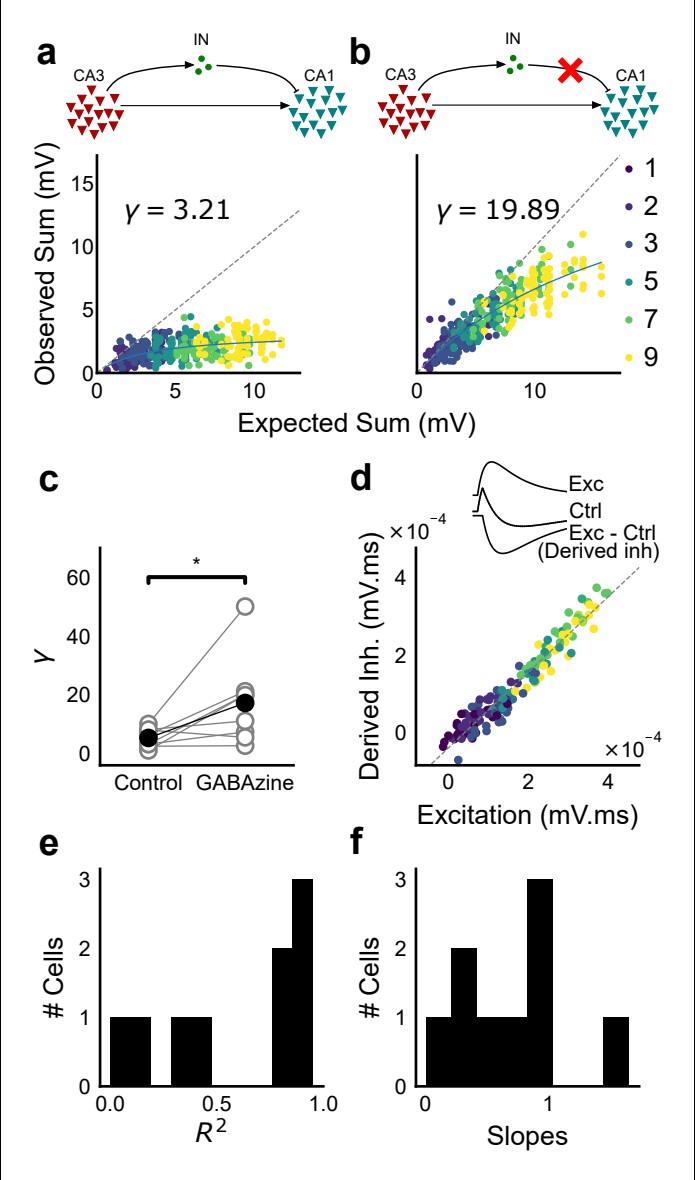

**Figure 5.** Blocking balanced inhibition at resting membrane potential attenuates SDN. (**a**) Top, schematic of experiment condition. Bottom, a cell showing divisive normalization in control condition. (**b**) Top, schematic of experiment condition with feedforward inhibition blocked (2 uM GABAzine). Bottom, responses of the same cell with inhibition blocked. The responses are much closer to the linear summation line (dashed). The blue lines in **a**, **b** are the fits of the DN model. The value of γ of the fit increases when inhibition is blocked. (**c**) Parameter γ was larger with GABAzine in bath (Wilcoxon rank sum test, p<0.05, n = 8 cells), implying reduction in normalization with inhibition blocked. (**d**) Excitation versus derived inhibition for all points for the cell shown in **a** (area under the curve) (Slope = 0.97, r-square = 0.93, x-intercept = 3.75e-5 mV.ms). Proportionality was seen for all responses at resting membrane potential. Top, 'Derived inhibition' was calculated by subtracting control PSP from the excitatory (GABAzine) PSP for each stimulus combination. (**e,f**) $R^2$ (median = 0.8) and slope values (median = 0.7) for all cells (n = 8 cells), showing tight IPSP/EPSP proportionality, and slightly more excitation than inhibition at resting membrane potentials.

(***Figure 6—figure supplement 1f***). Thus, inhibition clamps down the rising EPSP at progressively earlier times, resulting in saturation of PSP amplitude when excitation is increased (***Figure 6c,d***, ***Figure 8***). In ***Figure 8a and b***, we show using a schematic, how SDN emerges when inhibitory onset changes as an inverse function of input strength.

We observed that we were also able to capture the initial linear regime observed in *Figure 4b* by using the inverse relationship of delay with excitation in this conductance model. This can be understood as follows: at small excitatory input amplitudes, the EI delay is so large that inhibition arrives too late to affect the peak EPSP. At higher stimulus amplitudes the output response is now subjected to earlier, and hence increasingly effective inhibition, thus flattening the output curve (Appendix 1, *Video 1*, *Figure 3—figure supplement 1a*, *Figure 6c*).

We then tested if SDN required both EI balance and dynamic EI delay. We obtained values for balanced $g_{inh}$ for each I/E ratio, and then shuffled the order of the balanced inhibitory vector with the excitation. This implied that the average I/E ratio was maintained over all stimuli, but not for individual stimuli. This shuffled set of inhibitory conductance with respect to excitation was used to calculate $V_{max}$ (*Figure 6—figure supplement 1b*). Similarly, we obtained inhibitory delay ($\delta_{inh}$) corresponding to each value of excitation from the dynamic delay curve in *Equation (4)* (*Figure 6e*). We then shuffled the order of delays, keeping excitation in the same order (*Figure 6—figure supplement 1c*). In both cases, SDN was strongly attenuated, implying that both EI balance and inverse scaling of inhibitory delay were necessary for SDN (*Figure 6—figure supplement 1b,c*, Supplementary *Equation (6) to (8)*). Further, we transformed the membrane current equation (*Equation (5)*) into the form which resembles divisive normalization equation (Appendix 1), and saw that in this form, $\gamma$ depends on the intrinsic properties of the neuron, and is modulated by delays and EI ratios.

Thus, our analysis of a conductance model suggests that SDN could be a general property of balanced feedforward networks, due to two characteristic features: EI balance and inhibitory kinetics. Each of these variables may be subject to plasticity and modulation to attain different amounts of normalization (*Figure 8c,d*, *Figure 8—figure supplement 1*).

## Stimulus information is encoded both in amplitude and time

We next asked if the temporally advancing inhibition (*Figure 6e–h*) affected PSP peak time with increase in stimulus strength. We calculated the slope of the PSP peak times against the expected axis in the presence (Control) and absence of inhibition (GABAzine) for a given cell. If inhibition cut into excitation and resulted in advancing of peak times with increasing stimulus strength, the slope of peak times would be negative, as shown in *Figure 7a*. Conversely, when inhibition is blocked, slope of peak times is not expected to change much. We saw that for all cells, slope of the peak time with inhibition intact was lower than the slope in the case with inhibition blocked (*Figure 7b*) (Wilcoxon Rank sum test (p=0.006), n = 8 cells).

What does SDN mean for information transmission in balanced networks? While SDN allowed the cell to integrate a large range of inputs before reaching spiking threshold, it also resulted in diminishing changes in PSP peaks at larger inputs (*Figure 4b*). This implied that information about the input was partially 'lost' from the PSP amplitude. However, PSP times to peak became shorter (*Figure 7a,b*), hence potentially encoding some information about the input in this time variable (*Figure 7f*, *Figure 8b*). In contrast, while the peak amplitudes seen with GABAzine predicted the input more reliably, peak times of EPSPs did not change much with input

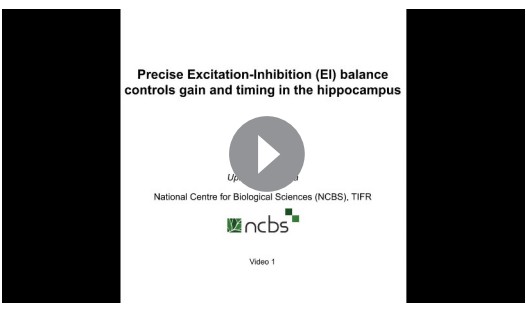

**Video 1.** Subthreshold divisive normalization emerges when onset delay of balanced inhibition dynamically decreases with excitation. (a) Schematic of the model of a single compartment neuron, which receives excitatory stimulus (in blue) at 20 ms, followed by an inhibitory stimulus (in orange) with variable onset delays. (b) Excitatory conductance (gluGbar) changes as shown in top most slider. Inhibitory conductance (I/E ratio*gluGbar) arrives after a dynamic or static delay. The orange and the blue dotted lines track the inhibition onset and the excitation peak, respectively. Their interaction point, marked by the orange dot, traces the relationship of excitatory conductance with dynamic or static delay. (c) EI summation plot (*Figures 3d* and *4b*) of PSP peak against excitation. Model shows SDN with dynamic EI delays, characterized by the initial linear zone followed by a sublinear zone for higher excitation values. SDN was lost when the EI delay was static. (d) Membrane voltage change as a result of only excitatory (dotted line), and integration of excitatory and inhibitory conductances (solid line) from panel **b**. Note how the peak time changes as a function of delays.
https://elifesciences.org/articles/43415#video1

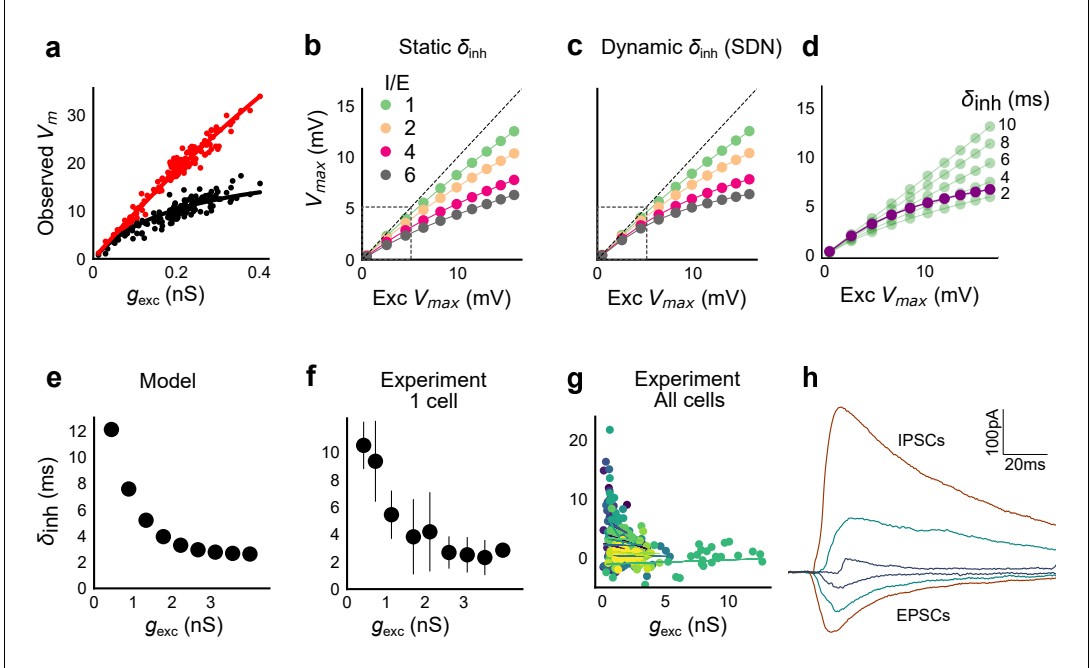

**Figure 6.** Conductance model predicts Excitatory-Inhibitory delay as an important parameter for divisive normalization. (**a**) Subthreshold responses from HH model, simulated with traces recorded from one voltage clamped cell (*Figure 2*). Non-linearly saturating curve, similar to SDN, obtained by simulating with both excitation and inhibition synaptic conductances (black), while the response profile is much more linear with only excitation (red). Each black point is the median response of an excitation trace paired with six different repeats of inhibition for that combination. (**b**) PSP peak amplitude with both excitatory and balanced inhibitory inputs is plotted against the EPSP peak amplitude with only excitatory input. Model showed sublinear behaviour approximating divisive inhibition for I/E proportionality ranging from 1 to 6 when the inhibitory delay was static. Different colours show I/E ratios. (**c**) Same as in b, except the inhibitory delay was varied inversely with excitatory conductance (as shown in **e**). Initial linear zone and diminishing changes in PSP amplitude, indicative of SDN were observed, and the normalization gain was sensitive to the I/E ratio. $\delta_{min}$= 2 ms, k = 0.5 nS$^{-1}$, and m = 8.15 ms. Note, the increased overlap in the initial zone (grey box) and the saturation of the PSP peaks in **c**, as compared to **b**. (**d**) Effect of changing EI delay, keeping I/E ratio constant (I/E ratio = 5). Divisive inhibition (green) seen while changing EI delay values from 2 to 10 ms. Divisive normalization (purple) emerges if delays are changed as shown in **e**. $\delta_{min}$= 2 ms, k = 0.5 nS$^{-1}$, and m = 8.15 ms. (**e**) Inverse relationship of EI delays with excitation. Inhibitory delay was varied with excitatory conductance in *Equation (4)* with $\delta_{min}$ = 2 ms, k = 2 nS$^{-1}$, and m = 13 ms. (**f**) Data from an example cell showing the relationship of EI delays with excitation. The relationship is similar to the prediction in **e**. Points are binned averages. Error bars are s.d. (**g**) Data from all cells showing delay as a function of excitation. Different colors indicate different cells (n = 13 cells). Grey lines are linear regression lines through individual cells. (**h**) Traces (from a voltage clamped neuron) showing the decreasing EI delay with increasing amplitude of PSCs. Each trace is an average of 6 repeats.

The online version of this article includes the following figure supplement(s) for figure 6:

**Figure supplement 1.** Sensitivity of SDN to EI balance and EI delay, and synaptic time courses used for model.

**Figure supplement 2.** HH model simulations with voltage clamped data show SDN.

(*Figure 7b,f*). Thus, PSP peak time may carry additional information about stimulus strength, when EI balance is maintained.

We quantified this using an information theoretical framework (*Shannon, 1948*). We took linear sum of 1-square PSP peak amplitudes (Expected sum), as a proxy for input strength. We then calculated the mutual information between Expected sum and PSP peak amplitudes of the corresponding N-squares, and between Expected sum and PSP peak timing (Materials and methods). Using this, we asked, how is the information about the input divided between PSP peak amplitude and timing? The total mutual information of both peak amplitude and peak timing with expected sum was slightly lesser in the presence of inhibition, but this difference was statistically not significant (*Figure 7e*) (Wilcoxon Rank sum test (<0.05), p=0.11, n = 7 cells). We found that peak timing had more information in presence of inhibition (control), and peak amplitude had more information in absence of inhibition (GABAzine) (Wilcoxon Rank sum test (<0.05), n = 7 cells) (*Figure 7f*). Further, we asked, how better can we predict the input, with the knowledge of peak timing, when the peak amplitude is already known? We found that in the presence of inhibition, peak amplitude

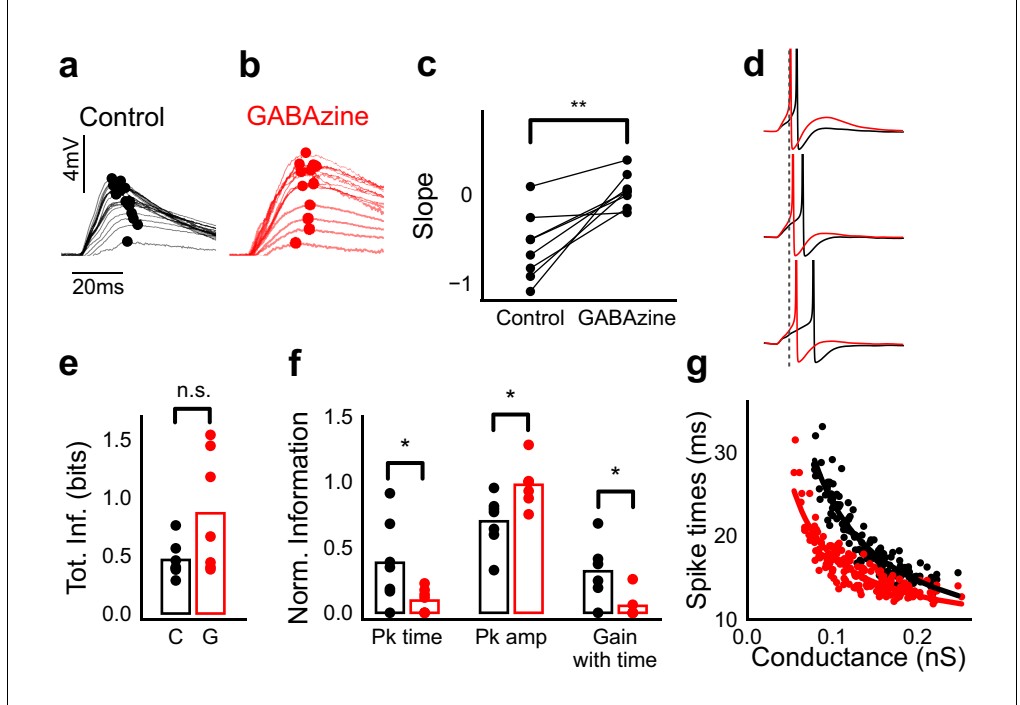

**Figure 7.** Advancing inhibitory onset changes PSP peak time and spike time with increase in stimulus strength. (**a,b**) The PSP peak arrived earlier following larger input in the control case (black), but not with GABAzine in bath (red). Traces for an example cell, binned (20 bins for Expected sum axis) and averaged, for control (black) and with GABAzine in bath (red). (**c**) Slope of the peak time was more negative in presence of inhibition (control) than when inhibition was blocked (GABAzine) (n = 8 cells). (**d**) Three example traces from the cell in **g** showing the relationship of spikes in presence (black) and absence of inhibition (red). Spikes were produced by HH model, using synaptic conductances from voltage clamp data. The separation between spike times of the two conditions increased with decrease in input conductance (top to bottom). (**e**) Total mutual information of peak amplitude and peak timing with expected sum was not significantly different between Control and GABAzine case (Wilcoxon Rank sum test (<0.05), p=0.11, n = 7 CA1 cells). (**f**) Normalized mutual information between Expected Vm and peak time, Expected Vm and peak amplitude, and conditional mutual information between Expected Vm and peak time, given the knowledge of peak amplitude. Normalized information was calculated by dividing mutual information by total information for each cell (as shown in **d**). Peak times carried more information in the presence of inhibition, and peak amplitudes carried more information in the absence of inhibition. There was higher gain in information about the input with timing if the inhibition was kept intact (Wilcoxon Rank sum test (p<0.05), n = 7 (Pk time, Pk amp) and (p=0.05) n = 6 (Gain with time) CA1 cells). (**g**) Relationship of spike time with excitatory conductance, in the presence (black) and absence of inhibition (red), for HH model simulations. All black points are medians of spikes of each excitation trace paired with six different repeats of inhibition for that combination.

The online version of this article includes the following figure supplement(s) for figure 7:

**Figure supplement 1.** Spike time changes with increasing input are steeper in presence of inhibition.

carried only a part of the total information about the input, and further knowledge of peak time substantially increased the total information. In contrast, in the absence of inhibition, peak amplitude carried most of the information about input, and there was very little gain in information with the knowledge of peak times (*Figure 7f*) (Wilcoxon Rank sum test (=0.05), n = 6 cells).

We then asked if the PSP peak time changes are also reflected in spike times. Since most of our stimuli elicited subthreshold responses, studying spiking required an artificial depolarization stimulus. From simulations we found that several parameters of the model (including resting membrane potential, membrane capacitance, synaptic conductances, EI ratio and delay, and spike threshold) could affect the mapping of subthreshold responses to spike timing, suggesting that this is a rich substrate for modulation. Keeping this caveat in mind, we tested the temporal profile of spikes with our model. We let the model cell spike in response to EI (similar to the Control condition) and only E (Gabazine condition). We observed that SDN translated to the spiking domain by encoding stronger stimulus amplitudes as shorter spike latencies, similar to the subthreshold responses. The presence of inhibition decreased the steepness of spike time with conductance (*Figure 7d,g*, *Figure 7—figure supplement 1*). The separation between the two conditions was sensitive to the exact value of threshold. At threshold close to resting potential, the separation was low, because the cell spiked

before the effect of inhibition set in. For a given threshold, a subset of the cells showed enough separation between conditions (*Figure 7d,g*, *Figure 7—figure supplement 1*) and this value could be tuned to obtain maximum separation for each cell.

Overall, these results suggest that with inhibition intact, input information is shared between amplitude and time, and knowledge of peak time and amplitude together contains more information about input than either of them alone.

### Modulation of gating with SDN

We next asked how the two basic parameters - I/E ratio and EI delay - modulated the degree of normalization and kinetics of the SDN curve (*Figure 8c,d*). Using our conductance model, we measured the normalization parameter $\gamma$ ($\alpha = 0$, $\beta = 1$, *Equation (1)*) for a range of values of I/E ratio and delays, and found that normalization increased systematically with increase in I/E ratio as well as with increase in the steepness of the EI delay relationship (*Figure 8c*). This implies that the degree of normalization of not only an entire neuron, but subsets of inputs to a neuron, could be dynamically altered by changing these parameters. In terms of gating, for a neuron with all inputs tightly balanced, any subset of inputs with reduction in I/E ratio will be gated 'on', corresponding to a condition of higher $\gamma$. Neurons can thus differentially gate and respond to specific inputs, while still retaining the capacity to respond to other input combinations.

## Discussion

This study describes two fundamental properties of the CA3-CA1 feedforward circuit: balanced excitation and inhibition from arbitrary presynaptic CA3 subsets, and an inverse relationship of excitatory-inhibitory delays with CA3 input amplitude. We used optogenetic photostimulation of CA3 with hundreds of unique stimulus combinations and observed precise EI balance at individual CA1 neurons for every input combination. Stronger stimuli from CA3 led to proportional increase in excitatory and inhibitory amplitudes at CA1, and a decrease in the delay with which inhibition arrived. Consequently, larger CA3 inputs had shorter inhibitory delays, which led to progressively smaller changes in CA1 membrane potential. We term this gain control mechanism Subthreshold Divisive Normalization (SDN). This reduction in inhibitory delay with stronger inputs contributes to a division of input strength coding between PSP amplitude and PSP timing.

### Precise balance in the hippocampus

Our findings demonstrate that precise EI balance is maintained by arbitrary combinations of neurons in the presynaptic network, despite the reduced nature of the slice preparation, with no intrinsic network dynamics. This reveals exceptional structure in the connectivity of the network. Theoretical analyses suggest that networks can achieve detailed balance with inhibitory Spike Timing Dependent Plasticity (iSTDP) rules (*Hennequin et al., 2017*; *Luz and Shamir, 2012*; *Vogels et al., 2011*). Such an iSTDP rule has been observed in the auditory cortex (*D'amour and Froemke, 2015*). Given that balance needs to be actively maintained (*Xue et al., 2014*), we suspect that similar plasticity rules (*Hennequin et al., 2017*) may also exist in the hippocampus.

Precisely balanced networks, with all input subsets balanced, are well suited for input gating (*Barron et al., 2017*; *Hennequin et al., 2017*). The finding that most silent CA1 cells can be converted to place cells for arbitrary locations predicts the existence of an input gating mechanism (*Lee et al., 2012*), but the nature of this mechanism remains unknown. One prediction of precise balance is that inputs for multiple potential place fields may be balanced, and hence place field activity is gated 'off'. Evoked depolarizations (*Lee et al., 2012*) or dendritic plateau potentials (*Bittner et al., 2015*; *Bittner et al., 2017*), which potentiate the subset of active synapses, that is, change the I/E ratio (*Grienberger et al., 2017*), can flip the gate 'on', thereby converting a silent cell to a place cell for that specific place field. This reasoning corroborates the observation of homogenous inhibition suppressing out-of-field heterogeneously tuned excitation (*Grienberger et al., 2017*), while providing a finer, synaptic scale view of the gating mechanism.

### EI delays and temporal coding

In several EI networks in the brain, inhibition is known to suppress excitation after a short time delay, leaving a 'window of opportunity' for spiking to occur (*Higley and Contreras, 2006*; *Pouille and*

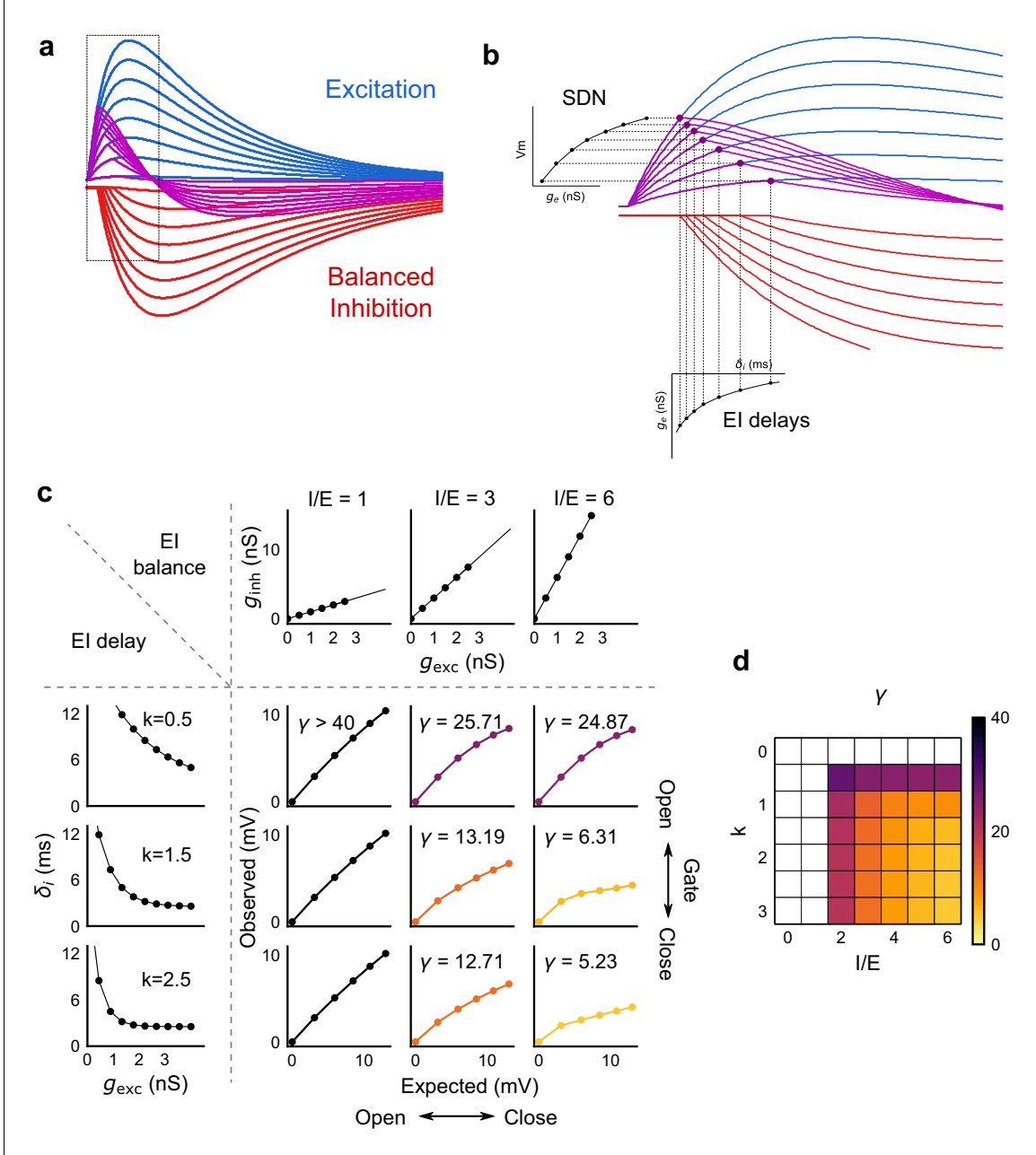

**Figure 8.** Emergence of SDN from balanced excitation and inhibition, coupled with dynamic EI delays. (**a**) Schematic showing precisely balanced EPSPs (blue) and corresponding IPSPs (red) summing to produce PSPs (purple). The EPSPs and IPSPs increase in equal input steps. (**b**) Zooming into the portion in the rectangle in **a**. Excitation onset is constant, but inhibition onset changes as an inverse function of input or conductance ($g_{exc}$), as shown in *Figure 6*. With increasing input, inhibition arrives earlier and cuts into excitation earlier for each input step. This results in smaller differences in excitatory peaks with each input step, resulting in SDN. The timing of PSP peaks (purple) becomes progressively advanced, whereas the timing of EPSP peaks (blue) does not, consistent with our results in *Figure 7*. (**c,d**) Normalization as a function of the two building blocks – EI balance (I/E ratio) and EI delays (interneuron recruitment kinetics, k, as predicted by the model. Larger values of both imply greater normalization and increased gating. Colors of the SDN curves depict the value of gamma ($\gamma$), as shown in the phase plot in **d**. White squares are values of $\gamma$ larger than 40, where almost no normalization occurs.

The online version of this article includes the following figure supplement(s) for figure 8:

**Figure supplement 1.** PSP traces showing the effect of I/E ratio and inhibitory recruitment kinetics (k) on SDN.

*Scanziani, 2001*; *Wehr and Zador, 2003*). We have shown that balanced inhibitory input arrives

with a delay modulated by the excitatory input in a feedforward circuit. This inverse relationship of EI delay with excitation has not been explicitly shown, although *Heiss et al. (2008)* report a decrease in EI delays with increase in whisker stimulation speed in layer 4 cells. We show that modulation of EI delay by excitation helps encode the input information in both amplitude and timing of the PSP (*Figure 7*). Thus, large inputs could be represented with fewer spikes, while conserving input strength information in spike timing. In CA1, a classic example of such dual coding is theta phase precession (*Jensen and Lisman, 2000*). In addition, spike times during sharp wave ripples, gamma oscillations and time cell representations are also precise up to ~10 ms, which is the range of the dynamic 'window of opportunity' we observe. This dynamic window also implies that the neuron can transition from temporal integration mode at small input amplitudes to coincidence detection at large input amplitudes (*Gabernet et al., 2005*; *Higley and Contreras, 2006*; *Wehr and Zador, 2003*). Consistent with this range of spike-coding transformations, our simulations suggest that the precise mapping of subthreshold summation to spike timing can be effectively modulated by several cellular parameters as well as by details of input activity (*Figure 7d,g*).

## Subthreshold Divisive Normalization (SDN): a novel gain control mechanism

We have introduced Subthreshold Divisive normalization (SDN) as a novel gain control mechanism arising from EI balance and dynamic EI delays. Our study was uniquely able to observe SDN because of the large range of inputs possible (*Poirazi et al., 2003*) using patterned optical stimulation. While we observed no unidirectional correlation of the distance between input spots and their responses for most inputs (*Figure 4—figure supplement 1*), a limitation of this stimulation design is that some of the inputs may not be fully independent due physical proximity of stimulus spots. SDN expands the dynamic range of inputs that a neuron can accommodate before reaching spike threshold (*Figure 8—figure supplement 1b*). This is particularly useful for temporally coding, sparsely spiking neurons like CA1 (*Ahmed and Mehta, 2009*). So far, analogous gain control by divisive normalization has only been observed for firing rates of neurons (*Carandini and Heeger, 2011*). This implies that the timescales of gain change in DN are averaged over periods of tens of milliseconds, over which rates change. As opposed to this, in SDN, gain of every input is normalized at synaptic (millisecond) timescales. Our results add a layer of subthreshold gain control in single neurons, to the known suprathreshold gain control at the population level in CA1 (*Pouille et al., 2009*). This two-step gain control implies that the dynamic range of the population may be wider than previously estimated. While most experimental observations of firing rate gain change have been explained by the phenomenological divisive normalization equation, the mechanistic basis for normalization has been unclear. SDN provides a biophysical explanation for phenomenological divisive normalization by connecting EI ratios and delays with gain control.

I/E ratio can be changed by neuromodulation (*Froemke, 2015*; *Froemke et al., 2007*), by short term plasticity mechanisms (*Bartley and Dobrunz, 2015*; *Klyachko and Stevens, 2006*; *Tsodyks and Markram, 1997*) and by disinhibition (*Basu et al., 2016*). Although we show that EI delays are input amplitude dependent, they may also be modulated by external signals, or behavioural states such as attention (*Kim et al., 2016*) (*Figure 8c,d*). Such interneuron recruitment based changes have been shown to exist in thalamocortical neurons (*Gabernet et al., 2005*). Dynamic regulation of EI delay has been theoretically explored in balanced networks (*Bruno, 2011*; *Kremkow et al., 2010*) for temporal gating of transient inputs independently by amplitude and time. Thus, temporal gating by EI delays (*Kremkow et al., 2010*), combined with the amplitude gating by detailed balance (*Vogels and Abbott, 2009*) could be a powerful mechanism for gating signals (*Kremkow et al., 2010*) in the hippocampal feedforward microcircuit.

Several studies point toward the existence of precise EI balance in the cortex (*Atallah and Scanziani, 2009*; *Okun and Lampl, 2008*; *Wehr and Zador, 2003*; *Wilent and Contreras, 2005*; *Zhang et al., 2003*; *Zhou et al., 2014*), and here we have shown it in the hippocampus. We propose that input strength dependent inhibitory delay change may be a general property of feedforward network motifs. Together, these suggest that precisely balanced feedforward networks are elegantly suited for controlling gain, timing and gating at individual neurons in neural circuits.

# Materials and methods

**Key resources table**

| Reagent type (species) or resource | Designation | Source or reference | Identifiers | Additional information |
|---|---|---|---|---|
| Genetic reagent (*M. musculus*) | C57BL/6-Tg (Grik4-cre) G32-4Stl/J | Jackson Laboratory | Stock #: 006474 | Dr. Susumu Tonegawa's laboratory, MIT |
| Strain, strain background (Adeno-associated virus) | AAV5.CAGGS.Flex.ChR2-tdTomato.WPRE.SV40 | Penn Vector Core | | |
| Software, algorithm | MOOSE simulator | *Ray and Bhalla, 2008* | RRID:SCR_008031 | Dr. Upinder Bhalla's laboratory, NCBS |

## Animals

All experimental procedures were approved by the National Centre for Biological Sciences Institutional Animal Ethics Committee (Protocol number USB–19–1/2011), in accordance with the guidelines of the Government of India (animal facility CPCSEA registration number 109/1999/CPCSEA) and equivalent guidelines of the Society for Neuroscience. CA3-cre (*C57BL/6-Tg (Grik4-cre) G32-4Stl/J* mice, Stock number 006474) were obtained from Jackson Laboratories. The animals were housed in a temperature controlled environment with a 14 hr light: 10 hr dark cycle, with *ad libitum* food and water.

## Virus injections

21–30 days old male transgenic mice were injected with Lox-ChR2 (AAV5.CAGGS.Flex.ChR2-tdTomato.WPRE.SV40) virus obtained from University of Pennsylvania Vector Core. Injection coordinates used were −2.0 mm RC, ±1.9 mm ML, −1.5 mm DV. ~300–400 nl solution was injected into the CA3 region of left or right hemisphere with brief pressure pulses using Picospritzer-III (Parker-Hannifin, Cleveland, OH). Animals were allowed to recover for at least 4 weeks following surgery.

## Slice preparation

8–12 week (4–8 weeks post virus injection) old mice were anesthetized with halothane and decapitated post cervical dislocation. Hippocampus was dissected out and 350 um thick transverse slices were prepared. Slices (350 microns) were cut in ice-cold high sucrose artificial cerebro-spinal fluid (hsaCSF) containing (in mM): 87 NaCl, 2.5 KCl, 1.25 $NaH_2 PO_4$, 25 $NaHCO_3$, 75 sucrose, 0.5 $CaCl_2$, 7 $MgCl_2$. Slices were stored in a holding chamber, in artificialcerebro-spinal fluid (aCSF) containing (in mM) - 124 NaCl, 2.7 KCl, 2 $CaCl_2$, 1.3$MgCl_2$, 1.25 $NaH_2PO_4$, 26 $NaHCO_3$, and 10 glucose, saturated with 95% $O_2$/5% $CO_2$. After at least an hour of incubation, the slices were transferred to a recording chamber and perfused with aCSF at room temperature.

## Electrophysiology

Whole cell recording pipettes of 2-5MO were pulled from thick-walled borosilicate glass on a P-97 Flaming/Brown micropipette puller (Sutter Instrument, Novato, CA). Pipettes were filled with internal solution containing (in mM): 130 K-gluconate, 5NaCl, 10 HEPES, 1 $Na_4$-EGTA, 2 $MgCl_2$, 2 Mg-ATP, 0.5 Na-GTP and 10Phosphocreatinine, pH adjusted to 7.3, osmolarity ~285 mOsm. The membrane potential of CA1 cells was maintained near −65 mV, with current injection, if necessary. GABA-A currents were blocked with GABAzine (SR-95531, Sigma) at 2 uM concentration for some experiments. Cells were excluded from analysis if the input resistance changed by more than 25% (measured for 15/39 cells) or if membrane voltage changed more than 2.5 mV (measured for 39/39 cells, maximum current injected to hold the cell at the same voltage was ±15 pA) of the initial value. For voltage clamp recordings, the K-gluconate was replaced by equal concentration Cs-gluconate. Cells were voltage clamped at 0 mV (close to calculated excitation reversal) and −70 mV (calculated inhibition reversal) for IPSC and EPSC recordings respectively. At 0 mV a small component of APV sensitive

inward current was observed, and was not blocked during recordings. Cells were excluded if series resistance went above 25MO or if it changed more than 30% of the initial value, with mean series resistance being 15.7MO ± 4.5 MO s.d. (n = 13). For CA3 current clamp recordings, the cells were excluded if the $V_m$ changed by 5 mV of the initial value. For whole-cell recordings, neurons were visualized using infrared microscopy and differential interference contrast (DIC) optics on an upright Olympus BX61WI microscope (Olympus, Japan) fitted with a 40X (Olympus LUMPLFLN, 40XW), 0.8NA water immersion objective. Recordings were acquired on a HEKA EPC10 double plus amplifier (HEKA Electronik, Germany) and filtered 2.9 kHz and digitized at 20 kHz.

## Optical stimulation setup

Optical stimulation was done using DMD (Digital Micromirror Device) based Optoma W316 projector (60 Hz refresh rate) with its color wheel removed. Image from the projector was miniaturized using a Nikon 50 mm f/1.4D lens and formed at the focal plane of the tube lens, confocal to the sample plane. The white light from the projector was filtered using a blue filter (Edmund Optics, 52532), reflected off of a dichroic mirror (Q495LP, Chroma), integrated into the light path of the Olympus microscope, and focused on sample through a 40X objective. This arrangement covered a circular field of around 200 micron diameter on sample. 2.5 pixels measured one micron at sample through the 40X objective. Light intensity, measured using a power meter, was about 15 mW/mm$^2$ at sample surface. Background light from black screen usually elicited no or very little synaptic response at recorded CA1 cells. A shutter (NS15B, Uniblitz) was present in the optical path to prevent the slice from being stimulated by background light during the inter-trial interval. The shutter was used to deliver stimulus of 10–15 ms per trial. A photodiode was placed in the optical path after the shutter to record timestamps of the delivered stimuli.

## Patterned optical stimulation

Processing 2 was used for generating optical patterns. All stimuli were 16 micron squares subsampled from a grid. 16 micron was chosen since it is close to the size of a CA3 soma. The light intensity and square size were standardized to elicit typically one spike per cell per stimulus. The number of spikes varied to some extent based on the expression of ChR2, which varied from cell to cell. The switching of spots from one trial to next, at 3 s inter trial interval, prevented desensitization of ChR2 over successive trials (*Figure 1g*).

For a patched CA1 cell, the number of connected CA3 neurons stimulated per spot was estimated to be in the range of 0 to a maximum of 50 for responses ranging from 0 to 2 mV. These calculations assume a contribution of 0.2 mV per synapse (*Magee and Cook, 2000*) and release probability of ∼0.2 (*Murthy et al., 1997*). This number includes responses from passing axons, which could also get stimulated in our preparation.

We did not observe any significant cross stimulation of CA1 cells. CA1 cells were patched and the objective was shifted to the CA3 region of the slice, where the optical patterns were then projected. CA1 cells showed no response to optical stimulation because (i) Use of CA3-cre line restricted ChR2 to CA3 cells, (ii) physical shifting of the objective away from CA1 also made sure that any leaky expression, if present, did not elicit responses. Using a cre-based targeted optogenetic stimulation combined with patterned optical stimulation, we designed an experiment which was both more specific and more effective at exploring a large stimulus space. Unlike electrical stimulation, optical stimulation specifically excited CA3 pyramidal neurons, and hence the recorded inhibition was largely feedforward. We believe this specificity was crucial to the finding that I/E ratios for all stimuli to a cell are conserved. Electrical stimulation does not distinguish between neuronal subclasses, and in particular fails to separate out the inhibitory interneurons. Since a key part of our findings emerged from being able to establish a temporal sequence of activation of interneurons, it was crucial to exclude monosynaptic stimulation of interneurons in the experimental design. Second, patterned optical stimulation allowed us to explore a grid of 225 stimulus points in CA3, thereby obtaining a wide array of stimulus combination with large dynamic range, without compromising on the specificity of stimulation (*Figure 1*, *Figure 1—figure supplement 1*).

We used four different stimulus grids (*Figure 1—figure supplement 1*). All squares from a grid were presented individually (in random order) and in a stimulus set - randomly chosen combinations

of 2, 3, 5, 7, or 9, with 2, 3 or 6 repeats of each combination. The order of presentation of a given N square combination was randomized from cell to cell.

## Data analysis and code availability

All analyses were done using custom written software in Python 2.7.12 (numpy, scipy, matplotlib and other free libraries) and MatlabR2012b. All error bars are standard deviations. All analysis codes are available as a free library at (https://github.com/sahilm89/linearity; copy archived at https://github.com/elifesciences-publications/linearity).

## Pre-processing

PSPs and PSCs were filtered using a low-pass Bessel filter at 2 kHz, and baseline normalized using 100 ms before the optical stimulation time as the baseline period. Period of interest was marked as 100 ms from the beginning of optical stimulation, as it was the typical timescales of PSPs. Timing of optical stimulation was determined using timestamp from a photodiode responding to the light from the projector. Trials were flagged if the PSP in the interest period were indistinguishable from baseline period due to high noise, using a two sample KS test (p-value<0.05). Similarly, action potentials in the interest period were flagged and not analyzed, unless specifically mentioned.

## Feature extraction

A total of four measures were used for analyzing PSPs and PSCs (*Figure 3c*). These were mean, area under the curve, average and area to peak. This was done to be able to catch differences in integration at different timescales, as suggested by *Poirazi et al. (2003)*. Trials from CA1 were mapped back to the grid locations of CA3 stimulation for comparison of Expected and Observed responses. Grid coordinate-wise features were calculated by averaging all trials for a given grid coordinate.

## Subthreshold divisive normalization model

Different models of synaptic integration: Subtractive Inhibition, Divisive Inhibition, and Divisive Normalization models were obtained by constraining parameters in *Equation (1)*. The models were then fit to the current clamp dataset using lmfit. Reduced chi-squares (*Figure 4—figure supplement 2o*) and Bayesian Information Criterion (*Figure 4c*) were used to evaluate the goodness of fits of these models to experimental data.

## Single-compartment model

A single-compartment conductance-based model was created in Python using sympy and numpy. The model consisted of leak, excitatory and inhibitory synaptic conductances (*Equation (5)*, *Figure 6—figure supplement 1a*) to model the subthreshold responses by the CA1 neurons.

$$C_m \frac{dV_m}{dt} = g_{\text{leak}}(V_m - E_{\text{leak}}) + g_{\text{exc}}(V_m - E_{\text{exc}}) + g_{\text{inh}}(V_m - E_{\text{inh}}) \tag{5}$$

The parameters used for the model were taken directly from data, or literature (*Supplementary file 2*). The synaptic conductances $g_{\text{exc}}(t)$, and $g_{\text{inh}}(t)$ were modeled as difference of exponentials (*Equations (6) and (7)*):

$$g_{\text{exc}}(t) = \bar{g}_{\text{exc}} \left( \frac{e^{\left(\frac{-t}{\tau_{\text{decay}}}\right)} - e^{\left(\frac{-t}{\tau_{\text{rise}}}\right)}}{-\left(\frac{\tau_{\text{rise}}}{\tau_{\text{decay}}}\right)^{\frac{\tau_{\text{decay}}}{\tau_{\text{decay}} - \tau_{\text{rise}}}} + \left(\frac{\tau_{\text{rise}}}{\tau_{\text{decay}}}\right)^{\frac{\tau_{\text{rise}}}{\tau_{\text{decay}} - \tau_{\text{rise}}}}} \right) \tag{6}$$

$$g_{\text{inh}}(t) = \bar{g}_{\text{inh}} \left( \frac{e^{\left(\frac{\delta_{\text{inh}} - t}{\tau_{\text{decay}}}\right)} - e^{\left(\frac{\delta_{\text{inh}} - t}{\tau_{\text{rise}}}\right)}}{-\left(\frac{\tau_{\text{rise}}}{\tau_{\text{decay}}}\right)^{\frac{\tau_{\text{decay}}}{\tau_{\text{decay}} - \tau_{\text{rise}}}} + \left(\frac{\tau_{\text{rise}}}{\tau_{\text{decay}}}\right)^{\frac{\tau_{\text{rise}}}{\tau_{\text{decay}} - \tau_{\text{rise}}}}} \right) \tag{7}$$

For the divisive normalization case, the inhibitory delays ($\delta_{\text{inh}}$) were modeled to be an inverse function of $g_{\text{exc}}(t)$ (*Equation (4)*). In other cases, they were assumed to be constant and values were taken from *Supplementary file 2*.

## HH-based single-compartment model

A single-compartment Hodgkin Huxley model with parameters mentioned in *Supplementary file 3* was simulated in MOOSE (*Ray and Bhalla, 2008*) to analyze how measured synaptic conductances sum to cause CA1 somatic depolarization. To enable spiking, we included sodium and potassium delayed rectifier (KDR) channels in these neurons. Then, we drove this neuron with synaptic input as measured from voltage clamp data.

## Measurement of synaptic conductances

We calculated excitatory and inhibitory conductances using *Equation (5)*, while holding the neuron at inhibitory and excitatory reversal potentials respectively (*Zhou et al., 2014*, *Atallah and Scanziani, 2009*). To measure excitatory conductance ($g_{exc}$), we clamped the membrane to the inhibitory reversal potential ($E_{inh}$). In the absence of a stimulus, the holding current gave us the value of leak current ($I_{leak}$). Excitatory synaptic current ($I_{exc}$) was measured as the change in membrane current evoked by the input stimulus ($I_m - I_{leak}$), from the baseline of holding current. We calculated the $g_{exc}$ by dividing this stimulus evoked excitatory current by the excitatory driving force ($V_m - E_{exc}$). The same procedure was repeated at excitatory reversal to measure inhibitory conductance ($g_{inh}$) for each stimulus.

With this method, measurement of $g_{exc}$ and $g_{inh}$ at corresponding clamped membrane voltages was independent of the absolute value of $I_{leak}$. However, we needed to obtain an estimate of leak conductance ($g_{leak}$) for the purposes of the model (*Equation (5)*). We could not use the absolute value of $I_{leak}$ as measured in our voltage clamped neurons because of blockage of potassium channels with Cs internals. Hence, for use in our conductance model, $g_{leak}$ measurements were not taken from our voltage clamp data, and instead the value was taken from literature.

## Fitting data

Voltage clamp data was fit to a difference of exponential functions (*Equation (8)*, *Figure 6—figure supplement 1e*) by a non-linear least squares minimization algorithm using lmfit, a freely available curve fitting library for Python. Using this, we obtained amplitudes ($\bar{g}$), time course ($\tau_{rise}$, $\tau_{decay}$) and onset delay from stimulus ($\delta_{onset}$) for both excitatory and inhibitory currents. We then calculated inhibitory onset delay ($\delta_{inh}$) by subtracting onset delay of excitatory from inhibitory traces.

$$g(t) = \bar{g} \left( \frac{e^{\left(\frac{\delta_{onset}-t}{\tau_{decay}}\right)} - e^{\left(\frac{\delta_{onset}-t}{\tau_{rise}}\right)}}{-\left(\frac{\tau_{rise}}{\tau_{decay}}\right)^{\frac{\tau_{decay}}{\tau_{decay}-\tau_{rise}}} + \left(\frac{\tau_{rise}}{\tau_{decay}}\right)^{\frac{\tau_{rise}}{\tau_{decay}-\tau_{rise}}}} \right) \tag{8}$$

## Onset detection

Onsets were also detected using three methods. Since we propose onset delays to be a function of the excitation peak, we avoided onset finding methods such as time to 10% of peak, which rely on peaks of the PSCs. We used threshold based (time at which the PSC crossed a threshold), slope based (time at which the slope of the PSC onset was the steepest) and a running window based method. In the running window method, we ran a short window of 0.5 ms, and found the time point at which distributions of two consecutive windows became dissimilar, using a two sample KS test. Ideally, with no input, the noise distribution across two consecutive windows should remain identical. All three methods gave qualitatively similar results.

## Modeling detailed balanced synapses

Synaptic inputs were modeled as sums of probabilistically activated basal synapses with synaptic strengths taken from a lognormal distribution with shape and scale parameters as given by our one square current clamp data (shape = −0.39, scale = 0.80). The width of the weight distribution was altered by changing the scale parameter. Probabilistic synaptic activation was modeled as a binomial process, with synaptic 'release probability' for excitatory and inhibitory inputs set at 0.2 and 0.8, respectively.

Inhibitory inputs were generated with various degrees of correlation to the excitation, by shuffling the excitatory weights in differently sized bins, from one to the length of the excitatory weight vector, controlled by a parameter $\rho$. In this manner, as $\rho$ changed from 1 to 0, excitatory and inhibitory

weight vectors changed from paired (detailed balance) to completely unpaired but with identical mean and variance of the weight distributions (global balance).

These synapses could be engaged by delivering stimuli, with the number of synapses per stimulus sampled from a Poisson distribution with mean of 5 synapses per stimulus. The total number of excitatory and inhibitory synaptic inputs engaged by a stimulus were always identical. Each stimulus was repeated six times. The resultant means and standard deviations for excitatory and inhibitory inputs were plotted against each other to compare different degrees of correlation. The whole process was repeated 100 times, and correlations and r-squared values were averaged to generate the heatmaps.

## Mutual information calculation

Mutual information was calculated by non-parametric entropy estimation and histogram methods. NPEET (https://github.com/gregversteeg/NPEET) was used for non-parametric estimation of Mutual Information. The relationship between variables was shuffled 500 times to find the significance of the Mutual Information estimate. If the true value of MI was not larger than 90% of the distribution obtained by shuffling, mutual information was assumed to be 0. If the total information about the linear sum of one square responses using both peak amplitude and time could not be established with 90% confidence as described above, the cell was excluded from further analysis. We also used the histogram method to find the mutual information (data not shown), and saw a similar trend. Cells with fewer than 80 trials and less than 2 mV inter-quartile range in the linear sum from one square PSP were excluded from the analysis. The calculated linear sum from one square PSP peak amplitude responses, measured N-square peak amplitudes and time were binned with an equal number of bins. The number of bins were calculated using Sturges' Rule, which selects the number of bins as $1 + 3.3 \log n$, where n is the total number of observations for a given neuron. Bin frequencies were divided by the total number of responses to get the probability of occurrence p(x) of each bin.

Mutual Information was then calculated for all pairs of combinations between linear sum, peak amplitude and time using *Equation (9)* and *(10)*.

$$MI(X,Y) = H(X) + H(Y) - H(X,Y) \tag{9}$$

where Shannon's entropy $H(X)$ for a variable $X$, is given as:

$$H(X) = \sum_{x \epsilon X} -\mathrm{p(x)log2p(x)} \tag{10}$$

Further, conditional mutual Information was calculated to measure gain in information about input (linear sum) by knowledge of peak timing when peak amplitude is already known. It was calculated using *Equation 11*.

$$I(X;Y|Z) = H(X,Z) + H(Y,Z) - H(X,Y,Z) - H(Z) \tag{11}$$

Normalized mutual information was calculated by dividing mutual information between pairs of variables by the total information between all three variables (*Equation 12*).

$$I(X;Y,Z) = H(Z) + H(X,Y) - H(X,Y,Z) \tag{12}$$

## Cross-pulse adaptation

We individually presented five unique photostimulation spots in all possible pairwise combinations, with an inter-stimulus interval of 50 ms (*Dittman et al., 2000*), to test for the interaction using a Cross Pulse Adaptation protocol. We then compared the averages of ten repeats of the response for a given spot when it arrived second in the stimulus-pair, to when it came first. Hence, if there is facilitation caused due to the presence of the first spot, then we should observe that the response to the spot when it comes second is larger than when it comes first in the stimulus pair. To quantify this change, we calculated the ratio between the average response of the spot, when it arrives at the second place, to the response when it arrives at the first place. This gave us the Cross Pulse Ratio (*Figure 4—figure supplement 1b*). A necessary internal control was that the self-self spot pairs should get facilitated. However, we observed lack of facilitation for self-self pairs, for all the cells we tested (*Figure 4—figure supplement 1*, n = 6 cells). To ensure that this effect was not due

to a limitation of the preparation, we tested paired pulse facilitation with electrical stimulation on the same neuron which depressed with optical stimulation. We show that the neuron shows PPF with electrical, but not with optical stimulation (*Figure 4—figure supplement 1a*). Unlike electrical stimulation, which strongly and briefly stimulates many axonal fibres, optical stimulation targets neurons with varying degrees of strengths, and incomplete recovery of ChR2 from desensitization at such short timescales may be the reason for the second pulse not being as effective as the first one. This interfered with our ability to measure paired pulse facilitation and introduced uncertainty in interpreting cross-pulse effects. This precluded further investigation using this approach.

## Distributedness and physical distance between square patterns

We calculated the effect of the interaction due to physical proximity of photostimulation squares on the responses. We defined a quantity distributedness, as the sum of the distance between all simultaneously stimulated spots from the combined centre of mass of these spots (*Figure 4—figure supplement 1d*). We compared this to degree of sublinearity, that is the ratio between the Observed response (O) and the Expected sum (E) of individual squares. Thus, if the interaction between neighbouring squares caused sublinearity, we would see a positive correlation between the distributedness and O/E ratio (for the stimuli within an N-square set). Conversely, a negative correlation would imply supralinearity.

We also checked for any interaction that may be taking place between two different optical stimulation patterns. To quantify this, we measured distances on the grid map between all spots in all pairs of patterns, and compared it against the Vm change they caused at CA1 (*Figure 4—figure supplement 1e*).

## Acknowledgements

AB and SM were supported by NCBS/TIFR and Council of Scientific and Industrial Research (CSIR). We acknowledge support from the University Grants Commission/Israel Science Foundation grant (UGC/ISF No. F 6-18/2014(IC)). We acknowledge the National Mouse Resource facility funded by Department of Biotechnology for housing and maintaining all animals used in this study. We would like to thank Nikhila Krishnan and Shriya Palchaudhuri for help with genotyping; Ashesh Dhawale for help in building the optical stimulation setup; and, Sathyaa Subramaniyam, Deepanjali Dwivedi, Oliver Muthmann, Mehrab Modi, Dinesh Natesan, Aditya Gilra, Arvind Kumar and Rishikesh Narayanan for discussions and suggestions on the manuscript.

## Additional information

### Competing interests

Upinder Singh Bhalla: Reviewing editor, *eLife*. The other authors declare that no competing interests exist.

### Funding

| Funder | Grant reference number | Author |
| --- | --- | --- |
| University Grants Commission | UGC/ISF No. F 6-18/2014 (IC) | Upinder Singh Bhalla |
| Israel Science Foundation | UGC/ISF No. F 6-18/2014 (IC) | Upinder Singh Bhalla |
| Council of Scientific and Industrial Research | Senior Research Fellowship | Sahil Moza |
| National Centre for Biological Sciences | Graduate Student Fellowship | Aanchal Bhatia Sahil Moza |

The funders had no role in study design, data collection and interpretation, or the decision to submit the work for publication.

## Author contributions
Aanchal Bhatia, Conceptualization, Data curation, Formal analysis, Investigation, Visualization, Methodology, Writing—original draft, Writing—review and editing; Sahil Moza, Conceptualization, Data curation, Software, Formal analysis, Validation, Investigation, Visualization, Methodology, Writing—original draft, Writing—review and editing; Upinder Singh Bhalla, Conceptualization, Resources, Software, Supervision, Funding acquisition, Writing—original draft, Project administration, Writing—review and editing

## Author ORCIDs
Aanchal Bhatia (iD) https://orcid.org/0000-0003-4709-115X
Sahil Moza (iD) https://orcid.org/0000-0002-2225-8841
Upinder Singh Bhalla (iD) https://orcid.org/0000-0003-1722-5188

## Ethics
Animal experimentation: All experimental procedures were approved by the National Centre for Biological Sciences Institutional Animal Ethics Committee (Protocol number USB-19-1/2011), in accordance with the guidelines of the Government of India (animal facility CPCSEA registration number 109/1999/CPCSEA) and equivalent guidelines of the Society for Neuroscience. CA3-cre (C57BL/6-Tg (Grik4-cre) G32-4Stl/J mice, Stock number 006474) were obtained from Jackson Laboratories. The animals were housed in a temperature controlled environment with a 14-h light: 10h dark cycle, with ad libitum food and water.

## Decision letter and Author response
Decision letter https://doi.org/10.7554/eLife.43415.sa1
Author response https://doi.org/10.7554/eLife.43415.sa2

# Additional files
## Supplementary files
• Supplementary file 1. Table S1 Synaptic time courses chosen for the model. The median, 25% and 75% values for each of the four distributions in *Figure 6—figure supplement 1e* are shown.

• Supplementary file 2. Table S2 Parameters for the conductance model. Parameters for this model were either calculated using electrophysiological experimental conditions, taken from literature (Table S2a) or fit from data (Table S2b).

• Supplementary file 3. Table S3 Parameters for the HH based conductance model. Parameters for this model were either calculated using electrophysiological experimental conditions, or taken from literature. The simulations were conducted using synaptic conductances, measured from voltage clamp data (*Figure 2*).

• Transparent reporting form

## Data availability
All simulation data and code are open source and online, available at https://github.com/sahilm89/linearity (copy archived at https://github.com/elifesciences-publications/linearity). Data is available at Dryad (http://doi.org/10.5061/dryad.f456k4f).

The following dataset was generated:

| Author(s) | Year | Dataset title | Dataset URL | Database and Identifier |
|---|---|---|---|---|
| Aanchal Bhatia, Sahil Moza, Upinder Singh Bhalla | 2019 | Precise excitation inhibition balance controls gain and timing in the hippocampus | http://doi.org/10.5061/dryad.f456k4f | Dryad, 10.5061/dryad.f456k4f |

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

# Appendix 1

Here we compare the analytic form of the PSP peak with and without inhibition. This set of equations furthers our understanding of how the subthreshold divisive normalization takes effect, with changes in EI ratios and inhibitory delays. Here, $\omega$ represents the EI ratio at the time of postsynaptic depolarization peak, and $\eta$ represents the ratio of the excitatory conductances at peak depolarization time in the presence and absence of delayed inhibition.

Finding roots for *Equation 5*,

$$C_\mathrm{m}\frac{dV_\mathrm{m}}{dt} = g_\mathrm{leak} \cdot ( E_\mathrm{leak} - V_\mathrm{m}) + g_\mathrm{exc}(t) \cdot ( E_\mathrm{exc} - V_\mathrm{m}) + g_\mathrm{inh}(t) \cdot ( E_\mathrm{inh} - V_\mathrm{m}) = 0$$

$$g_\mathrm{leak}E_\mathrm{leak} + g_\mathrm{exc}(t^*)E_\mathrm{exc} + g_\mathrm{inh}(t^*) E_\mathrm{inh} - V_\mathrm{m}( g_\mathrm{leak} + g_\mathrm{exc}(t^*) + g_\mathrm{inh}(t^*)) = 0 \tag{A1}$$

Here $t^*$ is the time of PSP peak.

$$V_\mathrm{m}(t^*) = \frac{g_\mathrm{leak}E_\mathrm{leak} + g_\mathrm{exc}(t^*)E_\mathrm{exc} + g_\mathrm{inh}(t^*) E_\mathrm{inh}}{g_\mathrm{leak} + g_\mathrm{exc}(t^*) + g_\mathrm{inh}(t^*)} \tag{A2}$$

Subtracting $E_\mathrm{leak}$ from both sides,

$$\theta = V_\mathrm{m}(t^*) - E_\mathrm{leak} = \frac{g_\mathrm{leak}E_\mathrm{leak} + g_\mathrm{exc}(t^*)E_\mathrm{exc} + g_\mathrm{inh}(t^*) E_\mathrm{inh}}{g_\mathrm{leak} + g_\mathrm{exc}(t^*) + g_\mathrm{inh}(t^*)} - E_\mathrm{leak} \tag{A3}$$

$$\theta = \frac{g_\mathrm{exc}(t^*)(E_\mathrm{exc} - E_\mathrm{leak}) + g_\mathrm{inh}(t^*) (E_\mathrm{inh} - E_\mathrm{leak})}{g_\mathrm{leak} + g_\mathrm{exc}(t^*) + g_\mathrm{inh}(t^*)} \tag{A4}$$

Similarly, with $\bar{g}_\mathrm{inh} = 0$ (no inhibition case), let $t^{**}$ be the time of peak.

$$\varepsilon = \frac{g_\mathrm{exc}(t^{**})(E_\mathrm{exc} - E_\mathrm{leak})}{g_\mathrm{leak} + g_\mathrm{exc}(t^{**})} \tag{A5}$$

Let $\eta = \frac{g_\mathrm{exc}(t^*)}{g_\mathrm{exc}(t^{**})}$, $\omega = \frac{g_\mathrm{inh}(t^*)}{g_\mathrm{exc}(t^*)}$, $\Delta E_\mathrm{exc} = (E_\mathrm{exc} - E_\mathrm{leak})$, and $\Delta E_\mathrm{inh} = (E_\mathrm{inh} - E_\mathrm{leak})$

Dividing 4 by 5, and replacing using terms above:

$$\frac{\theta}{\varepsilon} = \frac{\frac{g_\mathrm{exc}(t^*)\Delta E_\mathrm{exc}}{g_\mathrm{exc}(t^{**})\Delta E_\mathrm{exc}} + \frac{g_\mathrm{inh}(t^*)\Delta E_\mathrm{inh}}{g_\mathrm{exc}(t^{**})\Delta E_\mathrm{exc}}}{\frac{g_\mathrm{leak} + g_\mathrm{exc}(t^*) + g_\mathrm{inh}(t^*)}{g_\mathrm{leak} + g_\mathrm{exc}(t^{**})}}$$

$$\frac{\theta}{\varepsilon} = \frac{\eta + \omega\eta\frac{\Delta E_\mathrm{inh}}{\Delta E_\mathrm{exc}}}{\frac{g_\mathrm{leak}}{g_\mathrm{leak} + g_\mathrm{exc}(t^{**})} + (1 + \omega)\frac{g_\mathrm{exc}(t^*)}{g_\mathrm{leak} + g_\mathrm{exc}(t^{**})} \cdot \frac{g_\mathrm{exc}(t^{**})\Delta E_\mathrm{exc}}{g_\mathrm{exc}(t^{**})\Delta E_\mathrm{exc}}}$$

$$\frac{\theta}{\varepsilon} = \frac{\eta\left(1 + \omega\frac{\Delta E_\mathrm{inh}}{\Delta E_\mathrm{exc}}\right)}{\frac{g_\mathrm{leak}}{g_\mathrm{leak} + g_\mathrm{exc}(t^{**})} + \frac{(1 + \omega)}{\Delta E_\mathrm{exc}}\eta\varepsilon}$$

Multiplying the numerator and denominator by $\frac{\Delta E_\mathrm{exc}}{\eta(1 + \omega)}$

$$\frac{\theta}{\varepsilon} = \frac{\frac{\Delta E_{\text{exc}}\left(1 + \omega \frac{\Delta E_{\text{inh}}}{\Delta E_{\text{exc}}}\right)}{(1+\omega)}}{\frac{g_{\text{leak}} \cdot \Delta E_{\text{exc}}}{(g_{\text{leak}} + g_{\text{exc}}(t^{**}))(1+\omega)\eta} + \varepsilon}$$

$$\frac{\theta}{\varepsilon} = \frac{\frac{\Delta E_{\text{exc}}}{(1+\omega)} + \left(\frac{\omega}{1+\omega}\right)\Delta E_{\text{inh}}}{\frac{\Delta E_{\text{exc}}}{\left(1 + \frac{g_{\text{exc}}(t^{**})}{g_{\text{leak}}}\right)(1+\omega)\eta} + \varepsilon}$$

$$\theta = \frac{\lambda \varepsilon}{\gamma + \varepsilon} \tag{A6}$$

$$\lambda = \frac{\Delta E_{\text{exc}}}{(1 + \omega)}\left(1 + \omega \frac{\Delta E_{\text{inh}}}{\Delta E_{\text{exc}}}\right) \tag{A7}$$

$$\gamma = \frac{\Delta E_{\text{exc}}}{(1+\omega)(1+\Phi)\eta}, \quad where \ \Phi \frac{g_{exc}(t^{**})}{g_{\text{leak}}} \tag{A8}$$

When the delay between excitation and inhibition is large, $\eta$ approaches 1, and $\omega$ approaches 0, leading to $\theta$ approaching $\varepsilon$. This corresponds to the region where the input-output relationship is almost linear at low values of $\varepsilon$, and becomes increasingly sublinear as $\varepsilon$ increases. As the values of $\eta$ and $\omega$ increase, the value of $\gamma$ decreases, leading to increasing normalization.

