## [Decision Letter]

Thank you for submitting your article "Precise excitation-inhibition balance controls gain and timing in the hippocampus" for consideration by *eLife*. Your article has been reviewed by three peer reviewers, and the evaluation has been overseen by Ronald Calabrese as the Senior and Reviewing Editor. The following individual involved in review of your submission has agreed to reveal his identity: Carlos D Aizenman (Reviewer #3).

The reviewers have discussed the reviews with one another and the Reviewing Editor has drafted this decision to help you prepare a revised submission.

Summary:

In this paper, Bhatia, Moza and Bhalla present a very interesting and exciting study in which they used pattern illumination of hippocampal slices to study how CA1 neurons integrate inputs from CA3 and how E/I balance is determined by the strength of the stimulus. The experimental design and the logic behind these experiments is impressive. Pattern illumination was achieved by grid illumination, allowing presentation of a different number and combinations of light squares that were projected onto channelrhodopsin-2 (ChR2) expressing CA3 neurons while patching CA1 cells. The authors show that E/I balance of CA1 cells is maintained at a large range of stimulation strengths (i.e., for a variable number of light squares). Importantly, by wisely designing the experiment and the analysis they found that inhibition is a result of feedforward activation of interneurons and that the response is not evoked by recurrent activity. Then they examined the subthreshold response of the cells to different combinations of light squares and show that summation of these inputs is sublinear, exhibiting saturation as the expected (linear) response gets larger. They claim that such summation is an indication for divisive normalization. In the remaining part of the paper they use experimental and modeling approaches to suggest that sublinearity is caused by both the counterbalancing effect of inhibition and a dynamic shortening of the delay between excitation and inhibition as stimulus strength increases. Overall, this is an elegant study which potentially is very important for understanding the mechanisms and functions of E/I balance in neuronal networks.

Essential revisions:

There are concerns related both to the experimental controls and analysis of the data, and to the modeling part that should be addressed. Specifically, there are three major concerns, which must be addressed. The direct comments of the reviewers amplify these points; the major focus of the revision should be these three points, and not on minor differences in the way the reviewers have framed their concerns.

1) The reviewers need to be convinced that each stimulus square elicits an independent response when assessing SUBLINEARITY. This will necessitate new experiments showing this independence. This was done with whole cell recording experiments of CA3 cells and looking at the firing when stimulated with different light squares in order to see how big their 'receptive fields' are and they appear to be large (Figure 1). As reviewers 2 (comment 1) and 3 (comment 1) suggest this independence should be scrutinized by testing adaptation or using a paired pulse protocol from different locations. These experiments are very limited in scope and should be doable within the 2 month revision period.

2) There is concern that what is measure is synaptic current and not spiking activity of the CA1 neurons. Can experiments be performed to assess spiking? Perhaps in the slice spiking is suppressed limiting the ability to assess spiking. Maybe cell-attached recordings, or whole cell recordings with a depolarizing bias current or elevated external K^+^ would allow assessment of spiking? Such experiments should be attempted but if they cannot not be performed successfully, the authors should discuss the limitations of the analysis. Alternatively, they could address this issue by doing realistic simulations that include HH dynamics (equations).

3) Several concerns were advanced about the modeling and suggestions are given for tightening it up. One specific concern is that the modeling part is somewhat disconnected from the recorded data. We don't know if the authors did VC measurements using the same cells in which they recorded Vm and can extract *g_exc_, g_inh_* and *g*_leak_. If not, more experiments will be too much to ask, but the manuscript could be reorganized such that all the modeling part will appear following the experimental data which would reduce the need to base the model on actual data.

*Reviewer #1:*

In summary, the authors provide fresh evidence that precise balance is a guiding principle of brain function. It's not conclusive or final proof, but I do believe this study is a clever and publication-worthy experimental study that will further our knowledge on neural dynamics and certainly spawn new models in the ongoing back and forth between model and experiment in this topic. I would support publication.

I think the paper could be improved if the authors took a bit more didactic approach in explaining why they choose to perform specific experiments (for example, the reason for Figures 3 and 4 only becomes clear in Figure 5) and what the hypotheses are. Sometimes the Results section remains very technical, and the guiding motivation is not supplied. In particular, I think that the co-presentation of model and experiment, sometimes without specific mention (like in Figure 2G) is confusing, and I wonder if it would be helpful to present a figure with the general strategy and the expectations from the model before diving into the experiment. Another point that I wonder about is the connection between the sub-threshold behaviour shown here, and the link to the discussed is the super-threshold behaviour of spike propagation. There are no recordings of firing rate responses (current clamp) at all, and I wonder why

*Reviewer #2:*

Overall, this is an elegant study which potentially is very important for understanding the mechanisms and functions of E/I balance in neuronal networks. While I like this study and strongly support it, I have a few major comments, related both to the analysis of the data and to the modeling part. I believe that with proper revision it can be improved. As I claim below, their interpretation of the role of dynamic change in the delay of inhibition in divisive normalization might be correct, but the current analysis is not strong enough to fully support it.

1) While the authors state that their light stimulation may activate passing axons (or dendrites) they did not record the contribution of this effect to the sublinearity of the response. Although this effect is probably small, as supported by the gabazine experiments (Figure 5), this possibility should be checked (note that Gabazine had a negligible effect for a few cells in Figure 5C). Clearly, this issue has no effect on the results and conclusions of the first part of the study, demonstrating fixed balance under various combinations of squares. However, it may explain some of the sublinearity in the second part.

The authors need to add control experiments in which CA3 cells are recorded and find if cells fire only when they are directly stimulated by the light. There are a couple of additional ways to test this issue. One is to do adaptation experiments in which the response of CA1 cells to single light squares is compared to the response to same stimulus just after stimulating repetitively other pixels (using small ISI to cause substantial synaptic depression). In other words, to test stimulus specific adaptation. In case of stimulation of distinct inputs, interactions are not expected, and thus any saturation is likely to be caused by their proposed mechanisms. Another way is to restrict the analysis to non-bordering squares and ask if under this constraint summation becomes more linear or remains as shown.

2) Stimulation strength was altered by increasing the number of squares. We do not know if a similar effect also exists when the light intensity on a fixed illuminated area is increased. Unlike the elegant design in which different number of pixels were used, changes in light intensity do not allow the expected response to be calculated but such an experiment may also show that E/I balance is maintained, further generalizing the conclusions. At this state the authors can briefly discuss this issue.

3) Modeling issues: The model supports the conclusions. However, it can be improved by taking into account evoked synaptic conductances. Saturation of the response with increasing stimulation strength (i.e. sublinear summation) can be trivial under some conditions, but less in other cases. It all depends on the actual evoked synaptic conductances for each stimulus, the time constant of the cells and their resting potential. Saturation of the response (when plotting measured vs. predicted ∆V) is expected at steady state (i.e., dv/dt = 0) for strong E+I input. As stimulation strength increases the voltage approaches the combined reversal potential (which can be well below zero and slightly above r.p. for E+I). Whether or not the expected response saturates near its reversal potential depends not only on *g_exc_*and *g_inh_* but also on *g*_leak_ and *_V_*_leak_ (i.e. r.p.). The larger the leak conductance, the smaller the individual responses, resulting in more linear summation. I am not sure if the actual leak conductance was measured and used in the simulations.

4) Clearly, the reduction in the delay of inhibition as stimulation strength increases, is important and strongly suggests that inhibitory cells fire earlier than expected. As the number of stimuli increases, the delay between excitation and inhibition becomes shorter, indicating that inhibitory cells fired earlier and probably with higher probability causing (supra) non-linearity in summation of inhibition. It is not clear if this was introduced into the model and if at all. Again, measurements of conductances could help here. Clearly higher *g_inh_* than expected can lead by itself to sublinear response and divisive effect. In fact, the same issue can be also applied to *g_exc_*. Hence, without knowing the individual evoked *g_inh_* and *g_exc_* for each cell and stimulation, the current modeling part is not strong enough.

5) A compressive approach to solve this issues is to simulate the response using measured *g*_leak_, *V*_leak_ and *g_exc_, g_inh_* (for individual responses and combined) and reconstruct Vm response when the delay is fixed (static) or when it is set as measured from the combined response (such simulations should also take into account the total capacitance of the cells, which can be measured from the time constant of the cells). I don't know if the authors have such data but if they do, the study can be clearly improved by these computations. This more realistic model, in contrast to the current model where some of the parameters were taken from the literature, will provide better understanding of the integration process in their experiments. In other words, if possible, find how cells sum their excitatory and inhibitory conductances and then build a model in which summation of G's are linear and then transform the summed G's (*g*_leak_, *g_exc_(t)* and *g_inh_(t)*) into subthreshold responses based on r.p., time constant.

6) Also related: The first question that arises is whether or not the onset of excitation is also affected by the increasing strength of stimulation. If so, this is a clear indication for non-linearity in the summation of excitation or that multiple squares stimulated different processes (dendrites or axons) of the same neurons. Please check and report.

7) If I understand the results of Figure 6, they are based on Equation 5 which is clearly a first order differential equation. However, when inspecting the details of the model in Appendix 1, I see that the authors examined the steady state solution, which clearly is not applicable for their data. I got lost here.

*Reviewer #3:*

1) The authors make it seem like the different combinations of "spots" will lead to different presynaptic inputs being activated, but from the CA3 recordings in Figure 1, it is clear that a single CA3 cell can be directly activated by CHR2 from a fairly wide area almost encompassing the entire grid. Thus many of these combinations of spots may actually be targeting the same cell which could have a fairly extensive dendritic field. And thus these are not necessarily independent inputs in many cases. If they do want to make this claim, the authors should do some tests to see how much overlap there really is, for example using a paired pulse protocol from different locations. If the inputs are independent there should be no interaction between them, if they do overlap there should be some facilitation or depression. Ultimately it doesn't matter to their conclusion, since increasing postsynaptic responses does mean an increased number of presynaptic release sites, but they can't really claim they are truly independent inputs without testing it.

2) The idea that SDN serves as a way to normalize cell output is intriguing. However they aren't really measuring spiking in the postsynaptic cell which would really be an indication of actual output. If they want to strengthen this came, measuring some sort of input/output curve measuring postsynaptic spikes would be the real test of their hypothesis. This could even be done with loose cell-attached extracellular recordings to get better spike output.

---

## [Author Response]

Essential revisions:There are concerns related both to the experimental controls and analysis of the data, and to the modeling part that should be addressed. Specifically, there are three major concerns, which must be addressed. The direct comments of the reviewers amplify these points; the major focus of the revision should be these three points, and not on minor differences in the way the reviewers have framed their concerns.1) The reviewers need to be convinced that each stimulus square elicits an independent response when assessing SUBLINEARITY. This will necessitate new experiments showing this independence. This was done with whole cell recording experiments of CA3 cells and looking at the firing when stimulated with different light squares in order to see how big their 'receptive fields' are and they appear to be large (Figure 1). As reviewers 2 (comment 1) and 3 (comment 1) suggest this independence should be scrutinized by testing adaptation or using a paired pulse protocol from different locations. These experiments are very limited in scope and should be doable within the 2 month revision period.

We agree with the reviewers that the receptive fields of CA3 cells are larger than one square. While we have tried to take measures to restrict the spot size as much as possible, there is still some spread in the tissue. However, we would like to note that while a spread of a few spots can excite the cell, this kind of input independence may not be a necessity for linear summation. On the other hand, we agree that this interaction should be better quantified and have tested the degree of independence of the input squares in following four ways:

First, we tried to do the Cross Pulse Adaptation experiment as suggested.

We individually presented 5 unique photostimulation spots in all possible pairwise combinations, with an inter-stimulus interval of 50 ms (Dittman, et al., 2000), to test for the interaction using a Cross Pulse Adaptation protocol. We then compared the averages of ten repeats of the response for a given spot when it arrived second in the stimulus-pair, to when it came first. Hence, if there is facilitation caused due to the presence of the first spot, then we should observe that the response to the spot when it comes second is larger than when it comes first in the stimulus pair. To quantify this change, we calculated the ratio between the average response of the spot, when it arrives at the second place, to the response when it arrives at the first place. This gave us the Cross Pulse Ratio. A necessary internal control was that the self-self spot pairs should get facilitated. However, we observed lack of facilitation for self-self pairs, for all the cells we tested (Figure 4—figure supplement 2, n= 6 cells). To ensure that this effect was not due to a limitation of the preparation, we tested paired pulse facilitation with electrical stimulation on the same neuron which depressed with optical stimulation. We show that the neuron shows PPF with electrical, but not with optical stimulation (Figure 4—figure supplement 2). Unlike electrical stimulation, which strongly and briefly stimulates many axonal fibres, optical stimulation targets neurons with varying degrees of strengths, and incomplete recovery of ChR2 from desensitization at such short timescales may be the reason for the second pulse not being as effective as the first one.

This interfered with our ability to measure paired pulse facilitation and introduced uncertainty in interpreting cross-pulse effects. This precluded further investigation using this approach.

Second, we restricted our analysis to non-bordering squares and fit the subthreshold divisive normalization model and checked for the value of the normalization parameter (*γ*), as suggested by reviewer #2. The degree of sublinearity and the input-output curve remained unchanged, as indicated by the similarity in the values of DN parameter gamma (Figure 4—figure supplement 2C). This rules out the hypothesis that interactions between neighboring squares account for the observed sublinearity in the SDN curve.

Third, we looked at responses at CA1, by calculating the effect of the distributedness of squares on the responses. We defined distributedness as the sum of the distance between all simultaneously stimulated spots from the combined centre of mass of these spots (Figure 4—figure supplement 2D). We compared this to degree of sublinearity, i.e., the ratio between the Observed response (O) and the Expected sum (E) of individual squares. Thus, if the interaction between neighbouring squares caused sublinearity, we would see a positive correlation between the distributedness and O/E ratio (for the stimuli within an N-square set). Conversely, a negative correlation would imply supralinearity.

We found that the median correlation between distributedness and the O/E ratio was 0.09, showing that distances between grid squares did not have a unidirectional relationship with the extent of sublinearity.

Fourth, we also checked for any interaction that may be taking place between two different optical stimulation patterns. To quantify this, we measured distances on the grid map between all spots on all pairs of patterns, and compared it against the Vm change they caused at CA1. We again found close to no correlation (median correlation 0.02) between how close the patterns were on the grid to the measured voltage response of CA1 (Figure 4—figure supplement 2E).

With the above approaches, we conclude that there is no consistent effect of locations of photostimulation squares on the observed sublinearity. Our previous result (Figure 5), where we also record responses of all stimulus combinations in absence of inhibition (using GABAzine), serves as an internal control to account for the effect of location of squares (as correctly pointed out by reviewer #2). In summary, while the stimulation fields of squares have some overlap, the above analyses show that it does not affect summation unidirectionally.

Following the recommendation of reviewers #2 and #3, we have explicitly stated that there may be some interaction between squares, and added with the above analyses to the manuscript (Figure 4—figure supplement 2, Materials and methods).

Text added to manuscript:

“Since individual neurons may be targeted by more than one grid square (Figure 1B), individual spots are not completely independent and may interact, especially given the spread in the CA3 pyramidal neuronal arbour. Our analyses show that this interaction does not have a strong or unidirectional effect on the responses of the combinations of squares (Figure 4—figure supplement 2, Figure 5B, D).”

2) There is concern that what is measure is synaptic current and not spiking activity of the CA1 neurons. Can experiments be performed to assess spiking? Perhaps in the slice spiking is suppressed limiting the ability to assess spiking. Maybe cell-attached recordings, or whole cell recordings with a depolarizing bias current or elevated external K^+^ would allow assessment of spiking? Such experiments should be attempted but if they cannot not be performed successfully, the authors should discuss the limitations of the analysis. Alternatively, they could address this issue by doing realistic simulations that include HH dynamics (equations).

We appreciate the reviewers’ concern about translation of SDN to spiking. The technical reason for not assessing SDN in the spiking domain previously, was that our optical stimuli did not usually elicit spikes spontaneously (unlike electrical stimulation, which synchronously activates much larger number of neurons). Further, the neurons did not spike with isolated feedforward input from CA3, and hence artificially making the neurons spike may lead to fallacious conclusions about how SDN translates to the spiking domain. With this caveat in mind, we have tried to assess the spiking of the neurons by two means:

1) Using conductances recorded from the voltage clamped cells, we did realistic simulations with HH dynamics to check for differences in spiking behaviour with and without inhibition. We saw that the spike timing of the neurons followed a similar declining relationship as predicted by SDN mechanism, and as shown for the subthreshold peaks in Figure 7A. We added the following text to describe this analysis:

“We then asked if the PSP peak time changes are also reflected in spike times. Since most of our stimuli elicited subthreshold responses, studying spiking required an artificial depolarization stimulus. […] For a given threshold, a subset of the cells showed enough separation between conditions (Figure 7D, G, Figure 7—figure supplement 1) and this value could be tuned to obtain maximum separation for each cell.”

2) We also recorded from neurons in slices using ramp voltage timed to coincide with the optical stimuli to see if this could give rise to spiking patterns that exhibited the same temporal profile as the subthreshold responses. In some cases this worked, but we found that this happened in a narrow range and there were several complicating factors. Our preliminary data suggest that there is potentially a rich mapping from the subthreshold depolarizations to spiking patterns, and we felt that this was outside the scope of our already long paper.

In summary, we were able to address the reviewer concerns through simulations as suggested. We did attempt the suggested experiments but these did not yield a simple outcome.

3) Several concerns were advanced about the modeling and suggestions are given for tightening it up. One specific concern is that the modeling part is somewhat disconnected from the recorded data. We don't know if the authors did VC measurements using the same cells in which they recorded V_m_ and can extract g_exc_, g_inh_ and g_leak_. If not, more experiments will be too much to ask, but the manuscript could be reorganized such that all the modeling part will appear following the experimental data which would reduce the need to base the model on actual data.

In order to respond to points 2 and 3, we did realistic simulation with HH dynamics using synaptic inputs that were recorded from voltage clamp data. We used the responses of our voltage clamp data set to extract *g_exc_*and *g_inh_*from the same cell. Since the voltage clamp traces were used exactly as recorded, the EI delay was built into the input. We used *g_leak_, E_leak_* values from literature because our voltage clamp cells have Cs internal solutions, which blocks K channels and changes the leak potential of the cells. On simulating the model, we observed curves resembling SDN at subthreshold potentials (added as Figure 6A). Moreover, we assessed suprathreshold behaviour with the same data (response to Editor, point #2), and saw that SDN translates to the spiking domain by encoding stronger stimulus amplitudes as shorter spike latencies (Figure 7D, G, Figure 7—figure supplement 1). In conclusion, our new simulations intricately link the model to the recorded data.

Reviewer #1:In summary, the authors provide fresh evidence that precise balance is a guiding principle of brain function. It's not conclusive or final proof, but I do believe this study is a clever and publication-worthy experimental study that will further our knowledge on neural dynamics and certainly spawn new models in the ongoing back and forth between model and experiment in this topic. I would support publication.I think the paper could be improved if the authors took a bit more didactic approach in explaining why they choose to perform specific experiments (for example, the reason for Figures 3 and 4 only becomes clear in Figure 5) and what the hypotheses are. Sometimes the Results section remains very technical, and the guiding motivation is not supplied.

We thank the reviewer for pointing this out. To clarify, we have added the following paragraph in the beginning of the Results section, which will serve as a roadmap to explain the motivation for the chosen experiments.

“In our study, we first utilize and characterize an optical stimulation protocol for CA3 pyramidal neurons, and measure intracellular responses at CA1 pyramidal neurons (Figure 1). […] In Figure 8 we summarize the analysis and suggest how SDN could contribute to input gating in the hippocampus.”

In particular, I think that the co-presentation of model and experiment, sometimes without specific mention (like in Figure 2G) is confusing, and I wonder if it would be helpful to present a figure with the general strategy and the expectations from the model before diving into the experiment.

We thank the reviewer for pointing out the need for a figure to clarify the model in Figure 2. We now have an additional schematic illustrating the strategy used in the model, as Figure 2—figure supplement 2H.

Another point that I wonder about is the connection between the sub-threshold behaviour shown here, and the link to the discussed is the super-threshold behaviour of spike propagation. There are no recordings of firing rate responses (current clamp) at all, and I wonder why

We have also added to the paper simulations which look at suprathreshold behavior of neurons simulated with the conductances as measured experimentally (Figure 7). Please refer to point #2 of response to Editor for further details.

Reviewer #2:Overall, this is an elegant study which potentially is very important for understanding the mechanisms and functions of E/I balance in neuronal networks. While I like this study and strongly support it, I have a few major comments, related both to the analysis of the data and to the modeling part. I believe that with proper revision it can be improved. As I claim below, their interpretation of the role of dynamic change in the delay of inhibition in divisive normalization might be correct, but the current analysis is not strong enough to fully support it.1) While the authors state that their light stimulation may activate passing axons (or dendrites) they did not record the contribution of this effect to the sublinearity of the response. Although this effect is probably small, as supported by the gabazine experiments (Figure 5), this possibility should be checked (note that Gabazine had a negligible effect for a few cells in Figure 5C). Clearly, this issue has no effect on the results and conclusions of the first part of the study, demonstrating fixed balance under various combinations of squares. However, it may explain some of the sublinearity in the second part.The authors need to add control experiments in which CA3 cells are recorded and find if cells fire only when they are directly stimulated by the light. There are a couple of additional ways to test this issue. One is to do adaptation experiments in which the response of CA1 cells to single light squares is compared to the response to same stimulus just after stimulating repetitively other pixels (using small ISI to cause substantial synaptic depression). In other words, to test stimulus specific adaptation. In case of stimulation of distinct inputs, interactions are not expected, and thus any saturation is likely to be caused by their proposed mechanisms. Another way is to restrict the analysis to non-bordering squares and ask if under this constraint summation becomes more linear or remains as shown.

We have addressed this important concern in point #1 in the responses to the Editor.

2) Stimulation strength was altered by increasing the number of squares. We do not know if a similar effect also exists when the light intensity on a fixed illuminated area is increased. Unlike the elegant design in which different number of pixels were used, changes in light intensity do not allow the expected response to be calculated but such an experiment may also show that E/I balance is maintained, further generalizing the conclusions. At this state the authors can briefly discuss this issue.

We thank the reviewer for suggesting an alternate route to look for E/I balance. As previously mentioned in the manuscript (Patterned optical stimulation, Materials and methods), the stimulus parameters (like intensity, spot size) were standardized to limit the stimulus spread at CA3, and retain the responses at CA1 in the subthreshold domain, so as to not engage feedback inhibition (Figure 1 description of Results). This constraint was the main reason why we did not record at other intensities.

*3) Modeling issues: The model supports the conclusions. However, it can be improved by taking into account evoked synaptic conductances. Saturation of the response with increasing stimulation strength (i.e. sublinear summation) can be trivial under some conditions, but less in other cases. It all depends on the actual evoked synaptic conductances for each stimulus, the time constant of the cells and their resting potential. Saturation of the response (when plotting measured vs predicted* ∆*V) is expected at steady state (i.e., dv/dt = 0) for strong E+I input. As stimulation strength increases the voltage approaches the combined reversal potential (which can be well below zero and slightly above r.p. for E+I). Whether or not the expected response saturates near its reversal potential depends not only on g_exc_ and g_inh_ but also on g_leak_ and V_leak_ (i.e. r.p.). The larger the leak conductance, the smaller the individual responses, resulting in more linear summation. I am not sure if the actual leak conductance was measured and used in the simulations.*
4) Clearly, the reduction in the delay of inhibition as stimulation strength increases, is important and strongly suggests that inhibitory cells fire earlier than expected. As the number of stimuli increases, the delay between excitation and inhibition becomes shorter, indicating that inhibitory cells fired earlier and probably with higher probability causing (supra) non-linearity in summation of inhibition. It is not clear if this was introduced into the model and if at all. Again, measurements of conductances could help here. Clearly higher g_inh_ than expected can lead by itself to sublinear response and divisive effect. In fact, the same issue can be also applied to g_exc_. Hence, without knowing the individual evoked g_inh_ and g_exc_ for each cell and stimulation, the current modeling part is not strong enough.5) A compressive approach to solve this issues is to simulate the response using measured g_leak_, V_leak_ and g_exc_, g_inh_ (for individual responses and combined) and reconstruct Vm response when the delay is fixed (static) or when it is set as measured from the combined response (such simulations should also take into account the total capacitance of the cells, which can be measured from the time constant of the cells). I don't know if the authors have such data but if they do, the study can be clearly improved by these computations. This more realistic model, in contrast to the current model where some of the parameters were taken from the literature, will provide better understanding of the integration process in their experiments. In other words, if possible, find how cells sum their excitatory and inhibitory conductances and then build a model in which summation of G's are linear and then transform the summed G's (g_leak_, g_exc_(t) and g_inh_(t)) into subthreshold responses based on r.p., time constant.

Response to points 3-5 of reviewer #2:

We thank the reviewer for suggesting improvements to the model. We agree, and would like to point the reviewer to Appendix 1, Equations 6-8, which describe a set of the variables that go into SDN. We have also mentioned other potential sources of sublinearity in the Results section for Figure 5. In an HH model, we have reproduced SDN at membrane potential using the evoked EI traces recorded from all voltage clamped cells. For a detailed response, please refer to the points #2 and #3 of the Response to Editor.

6) Also related: The first question that arises is whether or not the onset of excitation is also affected by the increasing strength of stimulation. If so, this is a clear indication for non-linearity in the summation of excitation or that multiple squares stimulated different processes (dendrites or axons) of the same neurons. Please check and report.

For some cells, we observed that the onset of excitation may also shift with increasing stimulus strength. This is likely because a single cell may get stimulated by more than one squares. We have discussed the implication of the interaction of squares on sublinearity in the Response to Editor, point #1.

7) If I understand the results of Figure 6, they are based on Equation 5 which is clearly a first order differential equation. However, when inspecting the details of the model in Appendix 1, I see that the authors examined the steady state solution, which clearly is not applicable for their data. I got lost here.

Appendix 1 contains an analytic form of divisive normalization, looking at the inflexion point (PSP peak) with and without inhibition, when *g*_exc_ > 0; as opposed to the steady state. This is to understand the effect of participating variables and to show a mapping in form to the phenomenological divisive normalization equation (Equation 3). We have added description and reorganized text in the Appendix 1 for clarity.

“Appendix 1

Here we compare the analytic form of the PSP peak with and without inhibition. This set of equations furthers our understanding of how the subthreshold divisive normalization takes effect, with changes in EI ratios and inhibitory delays. Here, omega represents the EI ratio at the time of postsynaptic depolarization peak, and eta represents the ratio of the excitatory conductances at peak depolarization time in the presence and absence of delayed inhibition.”

Reviewer #3:1) The authors make it seem like the different combinations of "spots" will lead to different presynaptic inputs being activated, but from the CA3 recordings in Figure 1, it is clear that a single CA3 cell can be directly activated by CHR2 from a fairly wide area almost encompassing the entire grid. Thus many of these combinations of spots may actually be targeting the same cell which could have a fairly extensive dendritic field. And thus these are not necessarily independent inputs in many cases. If they do want to make this claim, the authors should do some tests to see how much overlap there really is, for example using a paired pulse protocol from different locations. If the inputs are independent there should be no interaction between them, if they do overlap there should be some facilitation or depression. Ultimately it doesn't matter to their conclusion, since increasing postsynaptic responses does mean an increased number of presynaptic release sites, but they can't really claim they are truly independent inputs without testing it.

We thank the reviewer for concisely pointing out two key points that can improve the paper. We have addressed these concerns. Please refer to the point #1 of the responses to the Editor.

2) The idea that SDN serves as a way to normalize cell output is intriguing. However they aren't really measuring spiking in the postsynaptic cell which would really be an indication of actual output. If they want to strengthen this came, measuring some sort of input/output curve measuring postsynaptic spikes would be the real test of their hypothesis. This could even be done with loose cell-attached extracellular recordings to get better spike output.

We have measured spike output due to subthreshold divisive normalization by simulation (Figure 7). Please refer to the point #2 of the responses to the Editor for a detailed response.